# Group-Level Data Selection for Efficient Pretraining

**Zichun Yu**[1]* **Fei Peng**[2] **Jie Lei**[2] **Arnold Overwijk**[2] **Wen-tau Yih**[2] **Chenyan Xiong**[1]

[1]Language Technologies Institute, Carnegie Mellon University
[2]Meta

## Abstract

The efficiency and quality of language model pretraining are largely determined by the way pretraining data are selected. In this paper, we introduce *Group-MATES*, an efficient group-level data selection approach to optimize the speed-quality frontier of language model pretraining. Specifically, Group-MATES parameterizes costly group-level selection with a relational data influence model. To train this model, we sample training trajectories of the language model and collect oracle data influences alongside. The relational data influence model approximates the oracle data influence by weighting individual influence with relationships among training data. To enable efficient selection with our relational data influence model, we partition the dataset into small clusters using relationship weights and select data within each cluster independently. Experiments on DCLM 400M-4x, 1B-1x, and 3B-1x show that Group-MATES achieves 3.5%-9.4% relative performance gains over random selection across 22 downstream tasks, nearly doubling the improvements achieved by state-of-the-art individual data selection baselines. Furthermore, Group-MATES reduces the number of tokens required to reach a certain downstream performance by up to $1.8\times$, substantially elevating the speed-quality frontier. Further analyses highlight the critical role of relationship weights in the relational data influence model and the effectiveness of our cluster-based inference. Our code is open-sourced at `https://github.com/facebookresearch/Group-MATES`.

## 1 Introduction

Improving the speed-quality frontier is essential for making large language models (LLMs) more efficient, scalable, and accessible across real-world applications [10, 15, 20]. Pretraining data selection [3, 25] provides a practical path to achieve that by identifying high-quality data [31, 44], optimizing domain mixtures [45], and constructing adaptive training curriculum [50]. Effective data selection approaches can nearly double the FLOPs-performance scaling efficiency of language models [11,

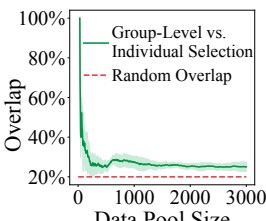
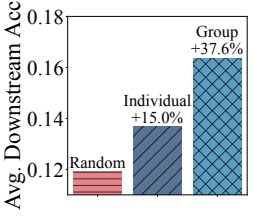

(a) Selected data overlap.     (b) Evaluation results.

Figure 1: Misalignment (a) and performance gap (b) between brute-force group and individual selection.

50], or enables smaller models to outperform larger counterparts [6].

Prevailing selection methods often evaluate the utility of each training data point *individually* [11], implicitly assuming that the overall utility of a data group is the sum of its elements. However, theoretical analyses [4, 34] reveal that the influence of a data group is shaped by complex interactions among data points rather than their isolated contributions. This discrepancy is particularly pronounced when selecting pretraining data. As shown in Figure 1a, in a typical pretraining data selection setting [25, 50], individual data selection quickly diverges from brute-force group selection after merely a hundred selected data points—less than a single batch in modern pretraining workflows.

---

*Part of the work done during an internship at Meta.

39th Conference on Neural Information Processing Systems (NeurIPS 2025).

In Figure 1b, training with group selection exhibits significantly higher downstream performance than individual selection, doubling the efficacy of pretraining data selection. Although group-level selection demonstrates tremendous potential, directly optimizing it is computationally prohibitive, which requires enumerating an exponential search space of all possible data subsets [38].

In this paper, we introduce *Group-MATES*, an efficient group-level data selection approach to optimize the speed-quality frontier of pretraining. Specifically, we parameterize costly group-level selection with a relational data influence model. To collect its training data, we sample group training trajectories of the language model and compute oracle data influences [50] alongside. The relational data influence model approximates the oracle data influence by weighting individual influence with relationships among training data. To enable efficient selection with our relational data influence model, we partition the dataset into small clusters using relationship weights and select data within each cluster, preserving essential relationships while reducing the computational cost.

We empirically verify the effectiveness of Group-MATES on DCLM [25], a standard pretraining data selection benchmark. DCLM is designed to assess the utility of data selection methods in enhancing pretraining, beyond the effects of basic cleaning and denoising that it already includes in data preprocessing. On DCLM 400M-4x, 1B-1x, and 3B-1x, Group-MATES achieves 3.5%-9.4% relative performance gains over random selection across 22 downstream tasks, nearly doubling the improvements achieved by state-of-the-art individual data selection baselines, including FineWeb-Edu Classifier [31], WebOrganizer [45], MATES [50], and Quad [51]. Furthermore, Group-MATES reduces the number of tokens required to reach a certain downstream performance by up to $1.8\times$ compared to random selection, substantially elevating the speed-quality frontier. Additional results confirm the effectiveness of Group-MATES in approaching group-level data selection and the necessity of having relationship weights in our relational data influence model. Further analyses validate that our cluster-based inference facilitates a more efficient data selection procedure while preserving crucial relational information.

We summarize the highlights of our work as follows:

1. We propose Group-MATES, a group-level data selection framework designed for efficient pretraining by parameterizing costly group selection with a relational data influence model.

2. We train our relational data influence model with sampled trajectories and enable its fast inference for data selection with influence-aware clustering.

3. Group-MATES sets a new state-of-the-art on DCLM and significantly elevates the speed-quality frontier. Further analyses highlight the essential role of relationship weights.

## 2   Related Work

Improving the speed-quality frontier is essential for making large language models (LLMs) more efficient, scalable, and accessible [10, 15, 20]. Pretraining data curation provides a practical path to achieve that by identifying and leveraging the most valuable data [3]. Standard approaches for data curation include: (1) *Domain reweighting* adjusts the mix of data from various sources (e.g., Wikipedia, GitHub) by determining optimal weights that work best for small proxy models [26, 45]. (2) *Synthetic data generation* employs generative models to rephrase [2, 28] or transform [49, 52] existing data, thereby augmenting or refining datasets. (3) *Data selection* encompasses various metrics to identify high-value data, ranging from rule-based filtering [30, 32], deduplication of semantically similar data [1, 35], proximity to high-quality corpora [25, 47], LLM-based quality scoring [31, 44], and data influence attribution [11, 13, 40, 50, 51]. The benefits of data selection are significant—recent techniques can double the speed-quality scaling of LLMs [31, 50], or enable smaller models to outperform larger counterparts trained on uncurated data [6].

The ideal goal of data selection is to identify the optimal subset of training data that maximizes model performance under resource constraints [3]. However, directly finding optimal subsets has been shown computationally prohibitive [12, 24], as it requires retraining the model on all possible subsets. To circumvent this challenge, a common assumption is that the most influential data points will also constitute the most influential subsets [11, 23]. Based on this, prior data selection methods primarily focus on evaluating the influence of individual data points [40, 50]. A typical approach for approximating individual data influence is influence function [22, 43], which utilizes first-order Taylor expansion to estimate how model parameters would change if a training point were infinitesimally

up-weighted. Beyond influence functions, DsDm [19] employs a linear regression model to estimate individual influences from subset training runs, while MATES [50] proposes a data influence model to parameterize individual influences. Both approaches have demonstrated notable success in improving the efficiency and effectiveness of pretraining.

While approximating group-level influences by individual influences can be computationally efficient, it often introduces substantial inaccuracies, as data points rarely contribute to model performance in isolation [17]. In particular, theoretical analyses [4, 34] show that group-level data influences contain relationship terms that cannot be captured by individual influences, while empirical studies [17, 18] reveal that interactions among data points can either cancel out or amplify individual influences. To mitigate this gap, ZAMinfluence [5] iteratively selects the most influential point to approximate the maximization of group-level influences, i.e., the greedy algorithm [29]. Building upon this work, researchers effectively applied group-level influences in data pruning [48], enhancing trustworthiness [7, 33, 39], LLM fine-tuning [14], and data selection [37, 38].

## 3 Preliminary

In this section, we first introduce the formulation of pretraining data selection and then standard approaches to evaluate oracle data influences. Finally, we empirically illustrate the gap between group-level and individual data influence oracles.

**Pretraining Data Selection.** Given a size-$N$ pretraining dataset $\mathcal{D}$ and a training budget of $n$ data points, data selection approach aims to to identify the optimal size-$n$ subset $\mathcal{D}^*_{(n)} \subset \mathcal{D}$ that yields the best pretrained model. In general, large-scale pretraining operates in a data-rich, compute-constrained regime, where the available data pool $\mathcal{D}$ is much larger than what can be used for training given practical computational budgets. Thus, data selection is typically performed *without replacement*.

Formally, the optimal size-$n$ training subset $\mathcal{D}^*_{(n)}$ is the set that minimizes the loss over a reference data $\mathcal{D}_r$ after optimizing the model $\mathcal{M}$ on $\mathcal{D}^*_{(n)}$:

$$\mathcal{D}^*_{(n)} = \underset{\mathcal{D}_{(n)}}{\arg\min} \ \mathcal{L}(\mathcal{D}_r \mid \mathcal{M}^*_{\mathcal{D}_{(n)}}) \tag{1}$$

$$= \underset{\mathcal{D}_{(n)}}{\arg\min} \ \mathbb{E}_{(x,y)\sim\mathcal{D}_r}\ell(y \mid x; \mathcal{M}^*_{\mathcal{D}_{(n)}}), \tag{2}$$

where $\mathcal{M}^*_{\mathcal{D}_{(n)}}$ denotes the model trained to converge on the data subset $\mathcal{D}_{(n)}$ using an optimizer like Adam [21] and $\ell$ denotes the function to compute the model loss on an input-output pair $(x, y)$. Prior works optimize $\mathcal{D}^*_{(n)}$ with retraining-based data influence oracles and their approximations.

**Retraining-Based Data Influence Oracles and Approximations.** The oracle group-level data influence of a subset $\mathcal{D}_{(n)}$ is normally quantified by leave-$n$-out retraining [4]. In particular, leave-$n$-out retraining evaluates the influence $\mathcal{I}$ of a subset $\mathcal{D}_{(n)}$ by measuring the difference in model performance when the subset is included in the training data versus excluded:

$$\mathcal{I}(\mathcal{M}^*_{\mathcal{D}}, \mathcal{D}_{(n)}) = \mathcal{L}(\mathcal{D}_r \mid \mathcal{M}^*_{\mathcal{D}}) - \mathcal{L}(\mathcal{D}_r \mid \mathcal{M}^*_{\mathcal{D}\setminus\mathcal{D}_{(n)}}), \tag{3}$$

While this approach accurately captures complex interactions among data points, it is computationally infeasible in practice, as it requires retraining the model from scratch for every possible subset.

To make group-level influence computation more tractable, prior works [11, 23, 50] approximate it by decomposing the group influence into the sum of leave-one-out oracle individual influences:

$$\mathcal{I}(\mathcal{M}^*_{\mathcal{D}}, \mathcal{D}_{(n)}) \approx \sum_{x_i \in \mathcal{D}_{(n)}} \mathcal{I}(\mathcal{M}^*_{\mathcal{D}}, x_i), \tag{4}$$

$$\text{where } \mathcal{I}(\mathcal{M}^*_{\mathcal{D}}, x_i) = \mathcal{L}(\mathcal{D}_r \mid \mathcal{M}^*_{\mathcal{D}}) - \mathcal{L}(\mathcal{D}_r \mid \mathcal{M}^*_{\mathcal{D}\setminus\{x_i\}}). \tag{5}$$

Instead of working with converged models $\mathcal{M}^*_{\mathcal{D}}$, MATES [50] introduces *local probing* to capture the dynamic nature of data influence as the model evolves during training. This technique calculates model-aware oracle data influence by applying a single gradient update to the current model $\mathcal{M}$ using data $x_i$ and measuring the change in reference loss before and after this one-step update:

$$\mathcal{I}(\mathcal{M}, x_i) = \mathcal{L}(\mathcal{D}_r \mid \mathcal{A}(\mathcal{M}, x_i)) - \mathcal{L}(\mathcal{D}_r \mid \mathcal{M}), \tag{6}$$

where $\mathcal{A}(\mathcal{M}, x_i)$ denotes the output of one-step optimization of model $\mathcal{M}$ on a data point $x_i$. The theoretical connection between Eq. 6 and influence functions can be found in Appendix A.1.

To efficiently calculate oracle individual data influences, MATES trains a parametric *data influence model* $\Theta^{\text{Indiv}}$ that learns to map data points to their oracle individual data influences:

$$\mathcal{I}(\mathcal{M}, x_i) \approx \Theta^{\text{Indiv}}(x_i) = \mathbf{w}_o \cdot \mathbf{h}_{x_i}, \qquad (7)$$

where $\mathbf{w}_o$ denotes the regression weight that transforms the last hidden representation $\mathbf{h}_{x_i}$ of a language model to the individual influence prediction.

**Empirical Gap between Group and Individual Data Influence Oracles.** Although the approximation in Eq. 4 makes group-level influence computation tractable, it can introduce substantial errors when estimating oracle group-level influences [23, 34]. Prior studies indicate that these errors stem from the approximation's neglect of interaction effects among data points within the group [17, 18].

To illustrate the gap between group-level and individual data influence oracles, we conduct an empirical study with the following setup: we utilize an intermediate checkpoint $\mathcal{M}$ (specifically, DCLM [25] 400M-4x baseline model at step 12,288) during pretraining and continue training it on the selected data for 100 steps, using the decay stage of the WSD scheduler [16]. We then evaluate the model's performance on 22 downstream tasks from DCLM. Further details can be found in Section 5.

We first select data that minimizes oracle individual influences, denoted as $\mathcal{D}_{(n)}^{\text{indiv}}$:

$$\mathcal{D}_{(n)}^{\text{indiv}} \leftarrow \arg\min{}_{x_i \in \mathcal{D}}^{(n)} \mathcal{I}(\mathcal{M}, x_i), \qquad (8)$$

where $\arg\min{}_{x_i \in \mathcal{D}}^{(n)}$ denotes the set of the data points with the lowest-$n$ oracle individual influences.

For group-level influences, direct optimization is infeasible as the search space grows exponentially with the dataset size. Instead, we formulate group-level selection as a sequential process [38], greedily selecting the data point with the minimum oracle individual influence at each step:

$$\text{For } t = 1, \ldots, n : x_t = \arg\min_{x_i \in \mathcal{D} \setminus \mathcal{D}_{(t-1)}^{\text{group}}} \mathcal{I}(\mathcal{M}_t, x_i), \qquad (9)$$

$$\mathcal{D}_{(t)}^{\text{group}} \leftarrow \mathcal{D}_{(t-1)}^{\text{group}} \cup \{x_t\}, \ \mathcal{M}_{t+1} = \mathcal{A}(\mathcal{M}_t, x_t), \qquad (10)$$

$$\text{starting from } \mathcal{D}_{(0)}^{\text{group}} = \emptyset, \mathcal{M}_1 = \mathcal{M}. \qquad (11)$$

Figure 1a shows that the overlap between $\mathcal{D}_{(n)}^{\text{indiv}}$ and $\mathcal{D}_{(n)}^{\text{group}}$ (i.e., $\frac{|\mathcal{D}_{(n)}^{\text{indiv}} \cap \mathcal{D}_{(n)}^{\text{group}}|}{n}$) decreases rapidly as $n$ increases. The overlap becomes close to random even for a small $n = 100$, indicating substantial divergence between the two selected sets. Furthermore, training $\mathcal{M}$ on $\mathcal{D}_{(n)}^{\text{group}}$ doubles the performance gain over random compared to $\mathcal{D}_{(n)}^{\text{indiv}}$, as illustrated in Figure 1b. This discrepancy aligns with previous theoretical findings [4, 34] and highlights the potential to approach group-level selection.

## 4 Methods

In this section, we introduce *Group-MATES*, a novel group-level data selection framework to advance the speed-quality frontier of pretraining. First, we propose a relational data influence model to parameterize costly group-level selection (§4.1). Next, we describe how to train (§4.2) and efficiently infer with the relational data influence model (§4.3). Our overall pipeline is illustrated in Figure 2.

### 4.1 Parametric Approximation of Group Selection with Relational Data Influence Model

As greedy group selection outlined in Eq. 10 requires brute-force computation of all oracle data influences, it is prohibitively costly to apply throughout the entire pretraining process. To address this challenge, we propose a relational data influence model $\Theta^{\text{rel}}$ to predict the oracle data influence of each candidate data point $x_t$ given the previously selected set $\mathcal{D}_{(t-1)}^{\text{rel}}$ and thus parameterize the group selection procedure as:

$$\text{For } t = 1, \ldots, n : x_t = \arg\min_{x_i \in \mathcal{D} \setminus \mathcal{D}_{(t-1)}^{\text{rel}}} \Theta^{\text{rel}}\big(x_i \mid \mathcal{D}_{(t-1)}^{\text{rel}}\big),$$

$$\mathcal{D}_{(t)}^{\text{rel}} \leftarrow \mathcal{D}_{(t-1)}^{\text{rel}} \cup \{x_t\}. \qquad (12)$$

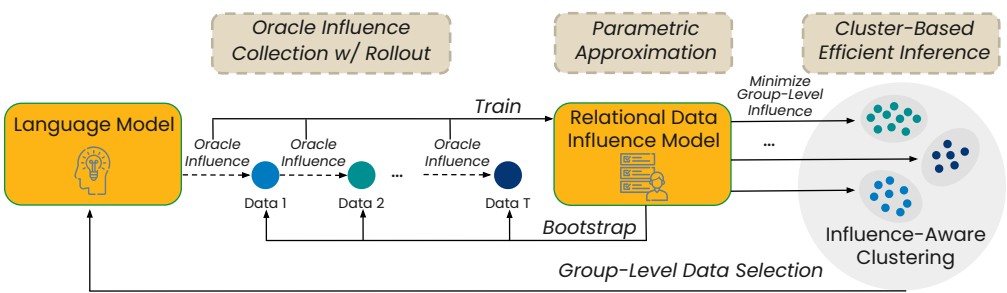

Figure 2: Overview of Group-MATES. We collect oracle data influences by sampling training data trajectories and train a relational data influence model to approximate them. This model then selects data that minimizes group-level influences within each influence-aware cluster.

Specifically, the prediction of the relational data influence model is formulated as the relationship-weighted individual data influence:

$$\mathcal{I}(\mathcal{M}_t, x_t) \approx \Theta^{\text{rel}}\big(x_t \mid \mathcal{D}_{(t-1)}\big) = \left[\alpha - \frac{\alpha}{\beta * (t-1)} \sum_{1 \leq i < t} R_{x_i, x_t}\right] * (\mathbf{w}_o \cdot \mathbf{h}_{x_t}), \tag{13}$$

$$\text{where } R_{x_i, x_t} = \text{sim}(\mathbf{h}_{x_i}, \mathbf{h}_{x_t}) \text{ is the relationship weight,} \tag{14}$$

$$\text{and } \mathbf{w}_o \cdot \mathbf{h}_{x_t} \text{ is the predicted individual data influence.} \tag{15}$$

$\alpha$ and $\beta$ are two trainable scaling factors initialized both from 1 and sim is the similarity function (e.g., cosine similarity) of two embeddings, ranging within $[-1, 1]$. The *relationship weight $R$* is designed to capture the interactive effects among training data points [4, 17]. We also provide a theoretical analysis in Appendix A.2 to demonstrate the inherent connection between our relationship weight and trajectory-specific influence functions [37].

## 4.2 Training Relational Data Influence Model with Rollouts

To gather supervision signals for training our relational data influence model, we use a rollout policy $\pi$ to sample training data trajectories $\mathcal{T}^\pi$ and collect oracle data influences $\mathcal{I}(\mathcal{M}_t, x_t)$ alongside:

$$\mathcal{T}^\pi \sim x_1 \dots x_{t-1} \xrightarrow{\pi\big(\cdot \mid \mathcal{D}_{(t-1)}\big)} x_t \dots x_T, \tag{16}$$

$$\text{where } \mathcal{D}_{(t)} \leftarrow \mathcal{D}_{(t-1)} \cup \{x_t\}, \ \mathcal{M}_{t+1} = \mathcal{A}(\mathcal{M}_t, x_t), \tag{17}$$

$$\mathcal{I}(\mathcal{M}_t, x_t) = \mathcal{L}(\mathcal{D}_r \mid \mathcal{M}_{t+1}) - \mathcal{L}(\mathcal{D}_r \mid \mathcal{M}_t). \tag{18}$$

where $T$ is the rollout length, a hyperparameter. We start with a random rollout policy $\pi_{\text{rand}}$ to train the initial data influence model $\Theta^{\text{rel}}_{\text{init}}$ by minimizing the mean squared error between its prediction $\Theta^{\text{rel}}\big(x_t \mid \mathcal{D}_{(t-1)}\big)$ and oracle data influence $\mathcal{I}(\mathcal{M}_t, x_t)$:

$$\Theta^{\text{rel}}_{\text{init}} = \arg\min_{\Theta^{\text{rel}}} \mathbb{E}_{\mathcal{T}^{\pi_{\text{rand}}}} \sum_{t=1}^{T} \left[\big(\Theta^{\text{rel}}\big(x_t \mid \mathcal{D}_{(t-1)}\big) - \mathcal{I}(\mathcal{M}_t, x_t)\big)^2\right]. \tag{19}$$

The distribution of oracle data influences is typically Gaussian [50], so random sampling primarily focuses on the mean of the distribution. To better approximate the full oracle distribution, we introduce *bootstrapping data influence model*, a targeted rollout policy $\pi_{\text{boot}}$ based on $\Theta^{*}_{\text{init}}$ that emphasizes the tail fractions of the oracle data influences. This policy explicitly samples data points corresponding to the lowest and highest predicted data influences:

$$\pi_{\text{boot}}\big(\cdot \mid \mathcal{D}_{(t-1)}\big) = \underbrace{\arg\min_{x_t \in \mathcal{D} \setminus \mathcal{D}_{(t-1)}}^{(K)} \Theta^{\text{rel}}_{\text{init}}\big(x_t \mid \mathcal{D}_{(t-1)}\big)}_{K \text{ lowest}} \cup \underbrace{\arg\max_{x_t \in \mathcal{D} \setminus \mathcal{D}_{(t-1)}}^{(K)} \Theta^{\text{rel}}_{\text{init}}\big(x_t \mid \mathcal{D}_{(t-1)}\big)}_{K \text{ highest}}, \tag{20}$$

where $K$ is the rollout width at every step. We then combine the supervision signals sampled from both $\pi_{\text{rand}}$ and $\pi_{\text{boot}}$ to train our final relational data influence model $\Theta^{\text{rel}}_{\text{final}}$:

$$\Theta^{\text{rel}}_{\text{final}} = \arg\min_{\Theta^{\text{rel}}} \mathbb{E}_{\mathcal{T}^{\pi_{\text{rand}}}, \mathcal{T}^{\pi_{\text{boot}}}} \sum_{t=1}^{T} \left[\big(\Theta^{\text{rel}}\big(x_t \mid \mathcal{D}_{(t-1)}\big) - \mathcal{I}(\mathcal{M}_t, x_t)\big)^2\right]. \tag{21}$$

### 4.3 Cluster-Based Efficient Inference of Relational Data Influence Model

Directly plugging $\Theta_{\text{final}}^{\text{rel}}$ into Eq. 12 can perform the group-level data selection. However, this iterative process can still be computationally intensive. To select a subset of size $n$, the selection involves $n$ steps. At each step $t$, we compute $\Theta_{\text{final}}^{\text{rel}}(x_i \mid \mathcal{D}_{(t)})$ for all $N - t$ remaining candidates from the original dataset of size $N$. This leads to a time complexity of $O(N \cdot n)$ calculation of relationship weights, which can be extremely slow for large datasets.

To speed up the selection, we propose a cluster-based inference approach that significantly reduces the number of relationship weight calculations. We first partition the selection pool $\mathcal{D}$ into $d$ clusters $\{\mathcal{C}^1, \mathcal{C}^2, \ldots, \mathcal{C}^d\}$. Consequently, Eq. 12 can be executed independently within each cluster, allocating the selection budget $n$ to each cluster proportionally to its size relative to the entire pool:

$$\text{For } i = 1, \ldots, d :$$
$$\text{For } t = 1, \ldots, \left\lceil n \cdot \frac{|\mathcal{C}^i|}{|\mathcal{D}|} \right\rceil : \mathcal{C}_{(t)}^i \leftarrow \mathcal{C}_{(t-1)}^i \cup \{ \operatorname*{arg\,min}_{x_j \in \mathcal{C}^i \setminus \mathcal{C}_{(t-1)}^i} \Theta_{\text{final}}^{\text{rel}}(x_j \mid \mathcal{C}_{(t-1)}^i) \}.$$
$$\mathcal{D}_{(n)} \leftarrow \bigcup_{i \in \{1, \ldots, d\}} \mathcal{C}_{\left(\left\lceil n \cdot \frac{|\mathcal{C}^i|}{|\mathcal{D}|} \right\rceil\right)}^i \tag{22}$$

This approach only computes relationship weights within each cluster rather than across the entire dataset. As a result, our cluster-based inference achieves a time complexity of $O\left(\frac{N \cdot n}{d}\right)$, enabling efficient group-level data selection for large-scale pretraining. In practice, running inference for each cluster independently with multiple threads can further reduce runtime.

Cluster-based inference can be viewed as an efficient approximation of brute-force inference in Eq. 12, which implicitly assumes independence between clusters and thus ignores relationships across them. To ensure that the most meaningful relationships are preserved during this process, we introduce *influence-aware clustering*. Specifically, our approach directly employs the relationship weight $R$ as the similarity metric for clustering, grouping data points with strong relationship weights into the same cluster. As a result, the relationship weights computed within each cluster closely approximate those computed over the full dataset.

Group-MATES is integrated into the pretraining pipeline in a model-aware manner [50]— pretraining is conducted in $S$ stages; after each stage $s$, we collect data influences with the current model $\mathcal{M}$, train the relational data influence model $\Theta^{\text{rel}}$, and utilize it to select training data for the next stage $s + 1$. This iterative process enables efficient, model-aware data selection throughout pretraining.

## 5 Experimental Setup

**Model and Data.** We conduct our main experiments following standard setups in DataComp-LM (DCLM) [25], a formalized competition to benchmark the effectiveness of pretraining data selection. The data curation pipeline in DCLM integrates heuristic cleaning, deduplication, and model-based filtering, yielding stronger baseline performance compared to other open-source datasets such as C4 [32], FineWeb [31], and RedPajama [41]. Beyond high data quality, DCLM also standardizes data loading, training hyperparameters, and evaluation tasks, making the competition strictly fair.

Specifically, we choose three experiment scales from DCLM, 400M-4x[2], 1B-1x, and 3B-1x. "400M" denotes the model size, and "4x" denotes the relative quantity of pretraining tokens for this model size, identified by the Chinchilla [15] optimum. We pretrain all models from scratch and evaluate pretrained models with 22 downstream tasks in either zero-shot or few-shot manners. These tasks provide a holistic assessment of the essential abilities of pretrained models, including commonsense reasoning, language understanding, reading comprehension, symbolic problem solving, and world knowledge. We use centered accuracy as the primary evaluation metric, where the accuracy per task is transformed to 0 when it equals random guessing and 1 corresponds to perfect accuracy. The average centered accuracy across all tasks is denoted as "Core score". A comprehensive list of the evaluation tasks is provided in Table 11.

---

[2]400M-4x is not a predefined setup in the original DCLM, but we extend its 400M-1x setup to train for 4x longer (4x more tokens) for better evaluation stability.

Table 1: Benchmarking different data selection methods on DCLM 400M-4x, 1B-1x, and 3B-1x settings. Dependencies on stronger LLMs (e.g., LLama3-70B-Instruct) are denoted by $*$. Best performances are marked **bold**.

| METHOD | COMMONSENSE REASONING (3 tasks) | LANGUAGE UNDERSTANDING (6 tasks) | READING COMPREHENSION (3 tasks) | SYMBOLIC PROBLEM SOLVING (5 tasks) | WORLD KNOWLEDGE (5 tasks) | CORE (22 tasks) |
|---|---|---|---|---|---|---|
| **400M-4X SETTING: 412M MODEL, 32.8B TOKENS** | | | | | | |
| EDU CLASSIFIER* | 0.29401 | 0.28287 | 0.03688 | 0.17480 | 0.24732 | 0.21821 |
| RANDOM | 0.25335 | 0.28315 | 0.10477 | 0.15643 | 0.22858 | 0.21356 |
| MATES | 0.28176 | 0.28358 | 0.14225 | 0.16296 | 0.22179 | 0.22260 |
| QUAD | **0.33437** | 0.27731 | 0.12080 | 0.15664 | 0.22124 | 0.22358 |
| GROUP-MATES | 0.29190 | **0.28735** | **0.14997** | **0.18890** | **0.22908** | **0.23362** |
| **1B-1X SETTING: 1.4B MODEL, 28.0B TOKENS** | | | | | | |
| EDU CLASSIFIER* | 0.33713 | 0.37612 | 0.14689 | 0.20967 | 0.33590 | 0.29257 |
| WEBORGANIZER* | 0.36042 | 0.39132 | 0.20225 | 0.18162 | 0.30865 | 0.29488 |
| RANDOM | 0.34994 | 0.38584 | 0.22059 | 0.18291 | 0.30784 | 0.29456 |
| MATES | 0.36331 | 0.39640 | 0.22548 | 0.19958 | 0.30415 | 0.30288 |
| QUAD | 0.34989 | **0.39913** | 0.16843 | 0.19864 | 0.30239 | 0.29340 |
| GROUP-MATES | **0.36997** | 0.39744 | **0.23922** | **0.20250** | **0.30793** | **0.30747** |
| **3B-1X SETTING: 2.8B MODEL, 55.9B TOKENS** | | | | | | |
| RANDOM | 0.44969 | 0.47816 | 0.27832 | 0.18070 | 0.37523 | 0.35603 |
| MATES | 0.44178 | 0.48263 | 0.30487 | 0.18497 | 0.37799 | 0.36139 |
| GROUP-MATES | **0.45874** | **0.48504** | **0.31094** | **0.19591** | **0.38146** | **0.36846** |

**Baselines.** We compare our method with (1) random selection (DCLM-Baseline); (2) edu classifier [31]: educational valuation of data distilled from LLama3-70B-Instruct [10]; (3) WebOrganizer [45]: domain construction with LLMs and mixture weight optimization via RegMix [26]. As WebOrganizer does not fully open-source their selection code, we copy their results in the same DCLM 1B-1x setup; (4) MATES [50]: data influence estimation with individual data influence models; and (5) Quad [51]: cluster-level influence estimation and diversification with multi-armed bandit [36]. These baselines cover state-of-the-art data selection techniques like LLM rating, domain mixtures, and individual data influence attribution. Some recent works, such as GREATS [38] and TSLOO [37], have not open-sourced their selection code for pretraining or evaluation results, hindering direct comparison. We also compare our method with earlier baselines (DSIR [47], SemDeDup [1], DsDm [11], and QuRating [44]) in Appendix C.4.

**Implementation Details.** We sample a size-128 subset from FLAN [42] as our reference data $\mathcal{D}_r$ for its exceptional generalization abilities [8]. FLAN does not overlap with our evaluation tasks. We initialize all parameters of our relational data influence model $\Theta^{rel}$ with `bge-base-en-v1.5` [46] except $\mathbf{w}_o$, which is randomly initialized. The similarity function for calculating relationship weight $R$ is cosine similarity, consistent with the original BGE design. We set the rollout length $T$=10 and rollout width $K$=5, and collect 20,000 rollout trajectories to train our relational data influence model. In inference, we partition all data points into $d$=10,000 clusters (the optimal choice in Quad) using k-means [27]. The number of pretraining stages $S$ is set to 2 and the selection ratio $\frac{n}{N}$ is set to 50%. More implementation details can be found in Appendix B. Ablations on key hyperparameters are presented in Appendix C.5, C.6, C.7, and C.8.

# 6 Evaluation Results

In this section, we present our main results on DCLM (§6.1). Then, we analyze the training of relational data influence models (§6.2), demonstrate the effectiveness of influence-aware clustering (§6.3), and finally present a case study (§6.4). Additional results can be found in Appendix C.

## 6.1 Main Results

**Overall Performance.** Table 1 summarizes the overall results on the DCLM benchmark. Group-MATES consistently outperforms random selection, achieving **3.5%–9.4%** relative improvements in Core scores across all three setups. Compared to our primary baseline, MATES, Group-MATES delivers superior performance on every subtask group, doubling its gain over random selection in the 400M-4x and 3B-1x settings. Notably, for the 3B-1x setup, Group-MATES maintains consistent improvements over random selection, whereas MATES exhibits diminishing returns, highlighting

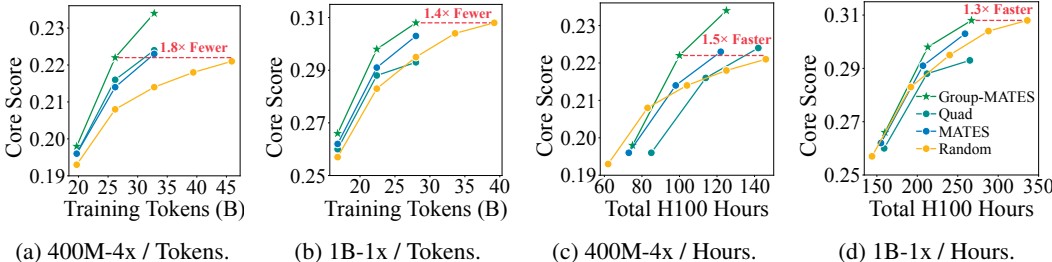

|  | (a) 400M-4x / Tokens. | (b) 1B-1x / Tokens. | (c) 400M-4x / Hours. | (d) 1B-1x / Hours. |

Figure 3: Core score comparison between Group-MATES and baselines w.r.t. pretraining tokens (a, b) and total H100 hours (c, d). Total H100 hours count both pretraining and data selection.

Table 2: Ablation on the key components in Group-MATES on DCLM 400M-4x setting.

| ABLATION | COMMONSENSE REASONING (3 tasks) | LANGUAGE UNDERSTANDING (6 tasks) | READING COMPREHENSION (3 tasks) | SYMBOLIC PROBLEM SOLVING (5 tasks) | WORLD KNOWLEDGE (5 tasks) | CORE (22 tasks) |
|---|---|---|---|---|---|---|
| GROUP-MATES | **0.29190** | **0.28735** | **0.14997** | **0.18890** | 0.22908 | **0.23362** |
| W/O RELATIONSHIP WEIGHT | 0.28074 | 0.28451 | 0.14301 | 0.17526 | 0.22951 | 0.22737 |
| W/O BOOTSTRAPPING | 0.28563 | 0.28139 | 0.14788 | 0.18304 | 0.22784 | 0.22924 |
| W/ SEMANTIC CLUSTERING | 0.28908 | 0.28172 | 0.14315 | 0.18524 | **0.23122** | 0.23042 |
| RANDOM | 0.25335 | 0.28315 | 0.10477 | 0.15643 | 0.22858 | 0.21356 |

the better scalability of group-level selection. These performance gains are substantial, as even the strong edu classifier—distilled from LLama3-70B-Instruct and recognized for its effectiveness on less curated datasets [31]—fails to outperform random selection in the 1B-1x setup. Edu classifier performs well on world knowledge tasks, since its selection strategy is optimized for educational value and thus favors knowledge-related data. To demonstrate the generalization ability of our method, we also conduct experiments on the C4 dataset in Appendix C.4, where Group-MATES consistently matches or surpasses MATES on **8 out of 9** evaluation tasks. In summary, Group-MATES demonstrates a significant advantage over individual data selection methods for pretraining, confirming the effectiveness of group-level data selection.

**Speed-Quality Frontier.** Figure 3 shows the evaluation results of Group-MATES and baselines with respect to pretraining tokens and total H100 hours. Token-based measurement reflects the compute cost of pretraining alone, as data selection can be easily parallelized with sufficient resources. Hour-based measurement accounts for both pretraining and data selection costs, representing the total compute used. In 400M-4x and 1B-1x settings, Group-MATES reduces the number of tokens by **1.8×** and **1.4×** needed to reach a given Core score compared to random selection. Furthermore, Group-MATES achieves **31.5%** and **20.5%** net efficiency gains (measured by H100 hours) in 400M-4x and 1B-1x setups, respectively. By contrast, MATES only achieves 16.4% and 10.1% net efficiency gains in 400M-4x and 1B-1x setups. Therefore, Group-MATES nearly doubles the net efficiency gain of MATES, further validating the advantage of our group-level selection over individual selection. These results demonstrate that Group-MATES substantially improves pretraining efficiency on the rigorous DCLM benchmark, achieving a state-of-the-art speed-quality frontier. A detailed breakdown of the compute overhead is provided in Appendix C.1, and the pretraining results under equalized compute are presented in Appendix C.2.

**Ablation Studies.** Table 2 shows the ablation studies of three key components in Group-MATES, namely relationship weight, bootstrapping data influence model, and influence-aware clustering. When we remove relationship weights during the selection process and consider only individual data influences, the performance gain over random selection drops by more than 30%, indicating that relational information plays a major role in the effectiveness of Group-MATES. In contrast, discarding the bootstrapping technique or replacing the influence-aware clustering with vanilla BGE semantic clustering also leads to noticeable performance degradation, though the magnitude of these drops is smaller compared to the effect of removing relationship weights. Overall, our ablation studies highlight the importance of incorporating relationship measurements between data points in our framework, extending the scope of data selection beyond the individual-level paradigm to account for inter-sample dependencies that better reflect the collective nature of the training data.

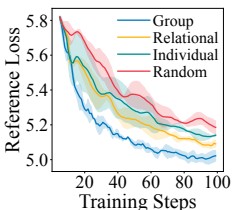 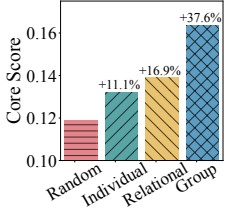 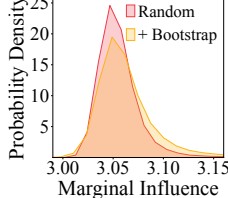 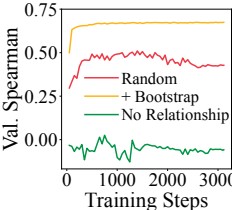

(a) Reference loss.  (b) Evaluation results.  (a) Influence distribution.  (b) Influence modeling.

Figure 4: Reference loss (a) and evaluation results (b) of greedy group-level selection, data selected by our relational data influence model, individual data influence model [50], and random.

Figure 5: Marginal influence distributions (a) and the performance of data influence models with random and bootstrap rollout policy, or random policy without relationship weight modeling (b).

**Comparison with Greedy Group-Level Selection.** This experiment compares the performance of our relational data influence model and the individual data influence model in MATES (Eq. 7) with greedy group-level selection, following the same experimental setup as Section 3. As shown in Figure 4a, the subset selected by our relational data influence model consistently achieves a lower reference loss than the individual one after the initial steps. The evaluation results in Figure 4b further validate the superiority of our relational data influence model, with a 5.8% relative performance gain compared to individual selection after the training. We also emphasize the significant potential of group-level selection, which nearly doubles the performance gain even in the short decay stage. Nevertheless, our method represents a critical step toward efficiently tackling group-level selection and has demonstrated its effectiveness.

## 6.2 Analyses on Relational Data Influence Model Training

This experiment analyzes the training of our relational data influence models. As shown in Figure 5a, using our bootstrapping rollout policy, the sampled oracle data influence distribution is more spread out. This demonstrates that bootstrapping effectively identifies more informative data points from the tails to train our relational data influence model. As a result, our relational data influence model with bootstrapping better approximates the oracle, improving the upper bound of validation Spearman correlation by 0.18 compared to the random rollout policy alone, as illustrated in Figure 5b.

We also demonstrate the necessity of having relationship weight in our relational data influence model. As shown in Figure 5b, when the relationship weight is removed from the model formulation (Eq. 13) and only individual influence is considered, the Spearman correlation drops significantly to near zero, indicating that the model fails to approximate the oracle data influence. This result suggests that the relationship weight is crucial for our relational data influence model to capture group-level influence, which aligns with previous theoretical findings [4, 34].

## 6.3 Effectiveness of Cluster-Based Inference

This experiment demonstrates the advantages of using influence-aware clustering for more efficient inference with relational data influence models. First, we compare the inference speed with clustering (Eq. 22) versus without clustering (Eq. 12). As shown in Figure 6a, cluster-based inference reduces inference time by several orders of magnitude, achieving over a $10^6 \times$ speedup compared to brute-force selection. This efficiency gain arises from computing relationship weights within clusters instead of across the entire dataset. Consequently, our data selection procedure effectively scales to large pretraining datasets containing millions of samples.

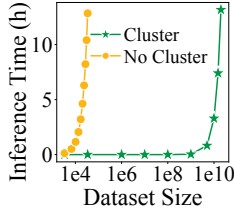 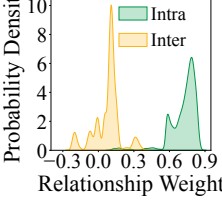

(a) Inference speedup.  (b) Relationship weight.

Figure 6: Inference speedup with clustering (a). Relationship weights in intra- and inter-cluster scenarios (b).

Table 3: Cancellation and amplification effects identified by our relational data influence model.

| Relation | Data 1 | Data 2 |
|---|---|---|
| Cancellation | Let the schools teach history, science, arts... hopefully allowing a greater degree of creativity and diversity to manifest. And parents should teach their children their philosophy, spiritual practices, and their wisdom as they see fit... | With technology, teachers are no longer going to be relevant, but on the contrary teachers are becoming more important, have very different role, of an expert, a manager and a facilitator... |
| Amplification | the object is to find integers x and z satisfying the Diophantine equation x-4z=44 A) Inasmuch as gcd A, 4) = 1 is a divisor of 44, there is a solution to this equation. Upon multiplying the relation 1 = 1 (-3) + 4 • 1 by 44 to get 44= l(-132) + 4-44... | Definition 2.2. Let a and b be given integers, with at least one of them different from zero. The greatest common divisor of a and b, denoted by gcd(a,b), is the positive integer d satisfying the following: (a) d | a and d | b. (b) If c | a and c | &, then c < d... |

To further validate that our influence-aware clustering can effectively approximate inference over the full dataset, we compare the distributions of the relationship weight $R$ (Eq. 14) between data points within the same cluster (intra-cluster) or across different clusters (inter-cluster). As shown in Figure 6b, the relationship weights are generally higher in the intra-cluster scenario, while inter-cluster relationship weights are distributed around 0. Therefore, our influence-aware clustering preserves essential relational information by keeping intra-cluster relationship weights in the inference procedure. This analysis validates that influence-aware clustering effectively approximates relationship computation over the full dataset by focusing on the stronger intra-cluster relationship weights.

## 6.4 Case Study

Finally, we present a case study in Table 3 to illustrate how to interpret the relationship weights given by our relational data influence model. Specifically, we analyze two representative examples of cancellation and amplification effects, which are identified when the relationship weight is significantly greater than 0 and less than 0, respectively. The cancellation effect in the first example arises from misaligned perspectives on education, where data 1 emphasizes parental influence and data 2 highlights teachers' critical roles. In contrast, the amplification effect in the second example emerges from complementary concepts: data 1 requires gcd for its problem solution, while data 2 provides a formal definition of gcd. Our study highlights the unique ability of our relational data influence model to capture complex interactions between training points, unlike semantic embedding models that focus solely on semantic similarity. We hope that our relational data influence model can serve as an analytic tool to discover and interpret more interesting interactions within pretraining data.

## 7 Conclusion

In this paper, we introduce Group-MATES, an efficient group-level data selection framework designed to optimize the speed-quality frontier of language model pretraining. On the DCLM benchmark, Group-MATES achieves 3.5%-9.4% relative performance gains over random selection, nearly doubles improvements from individual selection methods, and substantially reduces token and compute requirements for reaching target downstream performance levels. Further analyses show that modeling relationship weights is critical for accurately approximating oracle data influences.

Our work offers two key insights for data-centric pretraining. First, advanced data selection shall extend beyond the individual level, as model training is ultimately a collective effect. Second, modeling relationships among data provides a practical path toward scalable group-level data selection without extensive oracle data influence calculations. Future work can explore integrating Group-MATES into various training stages, extending relational modeling to further close the gap with greedy group selection, and investigating the theoretical limits of group-level selection efficiency. We hope that our work motivates a shift toward group-level perspectives in data selection, paving the way for more scalable and efficient foundation model pretraining.

**Acknowledgments**

We thank CMU Foundation and Language Model (FLAME) Center for computational support.

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

## A  Theoretical Analysis

### A.1  Connection between Oracle Data Influences and Influence Functions

We start from the oracle individual data influence, defined as the change in reference loss after locally probing the model $\mathcal{M}$ with $x_i$ [50]:

$$\mathcal{I}(\mathcal{M}, x_i) = \mathcal{L}(\mathcal{D}_r \mid \mathcal{A}(\mathcal{M}, x_i)) - \mathcal{L}(\mathcal{D}_r \mid \mathcal{M}), \tag{23}$$

where $\mathcal{A}(\mathcal{M}, x_i)$ denotes the model after training on $x_i$. To approximate this, we consider the optimal model $\mathcal{M}^*_{\epsilon, x_i}$ after upweighting $x_i$ by a small $\epsilon$:

$$\mathcal{M}^*_{\epsilon, x_i} = \arg\min_{\mathcal{M}} \frac{1}{n} \sum_{j=1}^{n} \mathcal{L}(x_j \mid \mathcal{M}) + \epsilon \mathcal{L}(x_i \mid \mathcal{M}), \tag{24}$$

and let $\mathcal{M}^* = \mathcal{M}^*_{0, x_i}$ be the original optimum. The influence function estimates the change in reference loss as $\epsilon \to 0$:

$$\mathcal{I}(\mathcal{M}, x_i) \approx \frac{d}{d\epsilon} \mathcal{L}(\mathcal{D}_r \mid \mathcal{M}^*_{\epsilon, x_i}) \Big|_{\epsilon=0} \tag{25}$$

Applying the chain rule,

$$\frac{d}{d\epsilon} \mathcal{L}(\mathcal{D}_r \mid \mathcal{M}^*_{\epsilon, x_i}) \Big|_{\epsilon=0} = \nabla_{\mathcal{M}} \mathcal{L}(\mathcal{D}_r \mid \mathcal{M}^*)^\top \frac{d\mathcal{M}^*_{\epsilon, x_i}}{d\epsilon} \Big|_{\epsilon=0} \tag{26}$$

To compute $\frac{d\mathcal{M}^*_{\epsilon, x_i}}{d\epsilon} \Big|_{\epsilon=0}$, we differentiate the optimality condition:

$$\nabla_{\mathcal{M}} \left[ \frac{1}{n} \sum_{j=1}^{n} \mathcal{L}(x_j \mid \mathcal{M}^*_{\epsilon, x_i}) + \epsilon \mathcal{L}(x_i \mid \mathcal{M}^*_{\epsilon, x_i}) \right] = 0 \tag{27}$$

Differentiating both sides with respect to $\epsilon$ and evaluating at $\epsilon = 0$ yields

$$H_{\mathcal{M}^*} \frac{d\mathcal{M}^*_{\epsilon, x_i}}{d\epsilon} \Big|_{\epsilon=0} + \nabla_{\mathcal{M}} \mathcal{L}(x_i \mid \mathcal{M}^*) = 0, \tag{28}$$

$$\frac{d\mathcal{M}^*_{\epsilon, x_i}}{d\epsilon} \Big|_{\epsilon=0} = -H^{-1}_{\mathcal{M}^*} \nabla_{\mathcal{M}} \mathcal{L}(x_i \mid \mathcal{M}^*), \tag{29}$$

where $H_{\mathcal{M}^*} = \frac{1}{n} \sum_{j=1}^{n} \nabla^2_{\mathcal{M}} \mathcal{L}(x_j \mid \mathcal{M}^*)$ is the Hessian. Substituting back to Eq. 26, we obtain the influence function approximation:

$$\mathcal{I}(\mathcal{M}, x_i) \approx -\nabla_{\mathcal{M}} \mathcal{L}(\mathcal{D}_r \mid \mathcal{M}^*)^\top H^{-1}_{\mathcal{M}^*} \nabla_{\mathcal{M}} \mathcal{L}(x_i \mid \mathcal{M}^*) \tag{30}$$

### A.2  Connection between Relationship Weight and Trajectory-Specific Influence Function

Trajectory-specific influence function [37] approximates a data point's conditional influence within the training trajectory, providing an intermediate approach to capture group-level influences. Formally, given a training batch sequence $\{\mathcal{B}_1, \mathcal{B}_2, \ldots, \mathcal{B}_T\}$, we estimate the influence of downweighting a training data point $x_i$ from batch $\mathcal{B}_t$ by a small amount $\epsilon$ on the reference loss $\mathcal{L}(\mathcal{D}_r \mid \mathcal{M}^*_{\epsilon, x_i}) - \mathcal{L}(\mathcal{D}_r \mid \mathcal{M}^*)$, where $\mathcal{M}^*$ and $\mathcal{M}^*_{\epsilon, x_i}$ are the final converged model state before and after the downweighting.

The model updates with Stochastic Gradient Descent (SGD) as:

$$\mathcal{M}_{t+1} = \mathcal{M}_t - \eta_t \sum_{x \in \mathcal{B}_t} \nabla_{\mathcal{M}} \mathcal{L}(x \mid \mathcal{M}_t) \tag{31}$$

Downweighting $x_i$ by $\epsilon$ modifies the update at step $t$:

$$\mathcal{M}_{t+1}(\epsilon) = \mathcal{M}_t - \eta_t \left( \sum_{x \in \mathcal{B}_t \setminus \{x_i\}} \nabla_{\mathcal{M}} \mathcal{L}(x \mid \mathcal{M}_t) + (1 - \epsilon) \nabla_{\mathcal{M}} \mathcal{L}(x_i \mid \mathcal{M}_t) \right) \tag{32}$$

The change in reference loss is approximated around $\epsilon = 0$ with first-order Taylor expansion:

$$\mathcal{L}(\mathcal{D}_r \mid \mathcal{M}^*_{\epsilon,x_i}) - \mathcal{L}(\mathcal{D}_r \mid \mathcal{M}^*) \approx \epsilon \cdot \left. \frac{\partial \mathcal{L}(\mathcal{D}_r \mid \mathcal{M}^*_{\epsilon,x_i})}{\partial \epsilon} \right|_{\epsilon=0} \tag{33}$$

$$\approx \epsilon \cdot \nabla_{\mathcal{M}} \mathcal{L}(\mathcal{D}_r \mid \mathcal{M}^*)^\top \left. \frac{\partial \mathcal{M}^*_{\epsilon,x_i}}{\partial \epsilon} \right|_{\epsilon=0} \tag{34}$$

Now we derive $\left. \frac{\partial \mathcal{M}^*_{\epsilon,x_i}}{\partial \epsilon} \right|_{\epsilon=0}$. At $t$, differentiating the modified update:

$$\left. \frac{\partial \mathcal{M}_{t+1}(\epsilon)}{\partial \epsilon} \right|_{\epsilon=0} = \eta_t \nabla_{\mathcal{M}} \mathcal{L}(x_i \mid \mathcal{M}_t) \tag{35}$$

For subsequent steps $j = t+1, \ldots, T-1,$:

$$\left. \frac{\partial \mathcal{M}_{j+1}(\epsilon)}{\partial \epsilon} \right|_{\epsilon=0} = \left. \frac{\partial \mathcal{M}_j(\epsilon)}{\partial \epsilon} \right|_{\epsilon=0} - \eta_j \sum_{x \in \mathcal{B}_j} \nabla_{\mathcal{M}}^2 \mathcal{L}(x \mid \mathcal{M}_j(\epsilon)) \left. \frac{\partial \mathcal{M}_j(\epsilon)}{\partial \epsilon} \right|_{\epsilon=0} \tag{36}$$

$$= (I - \eta_j H_j) \left. \frac{\partial \mathcal{M}_j(\epsilon)}{\partial \epsilon} \right|_{\epsilon=0} \tag{37}$$

Unrolling from $t+1$ to $T-1$:

$$\left. \frac{\partial \mathcal{M}^*_{\epsilon,x_i}}{\partial \epsilon} \right|_{\epsilon=0} = \left. \frac{\partial \mathcal{M}_T(\epsilon)}{\partial \epsilon} \right|_{\epsilon=0} = \left[ \prod_{j=t+1}^{T-1} (I - \eta_j H_j) \right] (\eta_t \nabla_{\mathcal{M}} \mathcal{L}(x_i \mid \mathcal{M}_t)) \tag{38}$$

Substituting into Eq. 34:

$$\mathcal{L}(\mathcal{D}_r \mid \mathcal{M}^*_{\epsilon,x_i}) - \mathcal{L}(\mathcal{D}_r \mid \mathcal{M}^*) \approx \epsilon \eta_t \nabla_{\mathcal{M}} \mathcal{L}(\mathcal{D}_r \mid \mathcal{M}^*)^\top \left[ \prod_{j=t+1}^{T-1} (I - \eta_j H_j) \right] \nabla_{\mathcal{M}} \mathcal{L}(x_i \mid \mathcal{M}_t) \tag{39}$$

When $x_i$ is totally removed from batch $\mathcal{B}_t$, i.e., $\epsilon = 1$:

$$\mathcal{L}(\mathcal{D}_r \mid \mathcal{M}^*_{\epsilon,x_i}) - \mathcal{L}(\mathcal{D}_r \mid \mathcal{M}^*) \approx \eta_t \nabla_{\mathcal{M}} \mathcal{L}(\mathcal{D}_r \mid \mathcal{M}^*)^\top \left[ \prod_{j=t+1}^{T-1} (I - \eta_j H_j) \right] \nabla_{\mathcal{M}} \mathcal{L}(x_i \mid \mathcal{M}_t) \tag{40}$$

Our relationship weight $R$ in Eq. 14 serves a similar purpose to $\prod_{j=t+1}^{T-1}(I - \eta_j H_j)$ by reweighting the influence of a data point based on its relationships with other points in the training trajectory.

# B  Experimental Details

Table 4: Training hyperparameters.

| HYPERPARAMETER | 400M-4X | 1B-1X | 3B-1X | RELATIONAL DATA INFLUENCE MODEL |
|---|---|---|---|---|
| STEPS | 31403 | 54923 | 107610 | 3086 |
| BATCH SIZE | 512 | 256 | 256 | 128 |
| SEQUENCE LENGTH | 2048 | 2048 | 2048 | 2048 (512 * 4) |
| MAX LEARNING RATE | 3E-3 | 3E-3 | 3E-3 | 5E-5 |
| OPTIMIZER | ADAMW | ADAMW | ADAMW | ADAMW |
| SCHEDULER | COSINE | COSINE | COSINE | COSINE |

**Language Model.** We pretrain all decoder-only language models from scratch following DCLM setups. The training employs cosine learning rate scheduler and AdamW optimizer [21]. All experiments are conducted on 8 GPUs, with detailed training hyperparameters provided in Table 4.

**Relational Data Influence Model.** Our relational data influence model is fine-tuned from `bge-base-en-v1.5` [46], which takes the last hidden state of the first token (i.e., [CLS]) as the

sentence embedding $\mathbf{h} \in \mathbb{R}^{768}$. As our base model only supports a maximum input sequence length of 512, but our pretraining sequence length extends to 2048, we split each sequence into four chunks and process them separately. The hidden states of four chunks are averaged to compute the final embedding $\mathbf{h}$. This vector is then multiplied by a regression weight $\mathbf{w}_o \in \mathbb{R}^{768}$ to predict individual influence $\mathbf{w}_o \cdot \mathbf{h}$. For relationship weights, the sim function is the cosine similarity between two embeddings, consistent with the original BGE design. The model is trained using the mean squared error loss between the predicted and Z-score normalized oracle data influences. The validation set consists of 1,000 sampled oracle influences. All training hyperparameters are listed in Table 4.

# C Additional Results

In this section, we provide a detailed compute breakdown (§C.1), results on different setups (§C.2, §C.3, §C.4), and extensive ablation studies (§C.5, §C.6, §C.7, §C.8) to support our findings.

## C.1 Compute Breakdown

Table 5: #FLOPs and H100 hours breakdown of Group-MATES.

| PROCESS | #FLOPs *1E19 | #FLOPs RATIO | H100 HOURS | HOURS RATIO |
|---|---|---|---|---|
| **400M-4X SETTING:** 412M MODEL, 32.8B TOKENS | | | | |
| MODEL PRETRAINING | 8.00 | 87.8% | 104.0 | 82.9% |
| ORACLE DATA INFLUENCE COLLECTION | 0.29 | 3.2% | 4.5 | 3.6% |
| DATA INFLUENCE MODEL TRAINING | 0.05 | 0.5% | 2.7 | 2.2% |
| DATA INFLUENCE MODEL INFERENCE | 0.77 | 8.5% | 14.2 | 11.3% |
| **TOTAL** | 9.11 | 100% | 125.4 | 100% |
| **1B-1X SETTING:** 1.4B MODEL, 28.0B TOKENS | | | | |
| MODEL PRETRAINING | 24.00 | 93.3% | 240.0 | 90.0% |
| ORACLE DATA INFLUENCE COLLECTION | 1.01 | 3.9% | 12.1 | 4.5% |
| DATA INFLUENCE MODEL TRAINING | 0.05 | 0.3% | 2.7 | 1.0% |
| DATA INFLUENCE MODEL INFERENCE | 0.65 | 2.5% | 12.0 | 4.5% |
| **TOTAL** | 25.71 | 100% | 266.8 | 100% |
| **3B-1X SETTING:** 2.8B MODEL, 55.9B TOKENS | | | | |
| MODEL PRETRAINING | 94.00 | 96.6% | 740.0 | 94.2% |
| ORACLE DATA INFLUENCE COLLECTION | 1.98 | 2.0% | 18.7 | 2.4% |
| DATA INFLUENCE MODEL TRAINING | 0.05 | 0.1% | 2.7 | 0.4% |
| DATA INFLUENCE MODEL INFERENCE | 1.30 | 1.3% | 23.9 | 3.0% |
| **TOTAL** | 97.33 | 100% | 785.3 | 100% |

We provide a detailed breakdown of compute used by Group-MATES in Table 5, measured either by #FLOPs or H100 hours. Notably, the data selection procedure of Group-MATES only accounts for 12.2%, 6.7%, and 3.4% of the total FLOPs in 400M-4x, 1B-1x, and 3B-1x setups, respectively. The relative selection cost in larger setups is generally smaller because their pretraining FLOPs dominate the total computation, while the training and inference costs of data influence model remain nearly stable. Considering the net gains achieved by Group-MATES, its compute overhead can be negligible.

## C.2 Equalized Compute Setup

To evaluate the effectiveness of Group-MATES under an equalized compute setup, we compare its performance against random selection using the same total FLOPs, as presented in Table 6. Although random selection utilizes more tokens for pretraining, Group-MATES consistently outperforms it across different scales. Specifically, Group-MATES achieves relative gains of 7.4%, 4.1%, and 3.0% in the 400M-4x, 1B-1x, and 3B-1x setups, respectively. These results highlight that merely increasing the number of training tokens does not yield comparable improvements to our selection method. Notably, the computational overhead of Group-MATES diminishes relative to the total pretraining cost as model and data scales increase. Furthermore, the selection process can be efficiently parallelized and decoupled from the pretraining. Our results underscore the scalability and efficiency of Group-MATES, making it an attractive preliminary step for large-scale pretraining.

We observe that in some sub-categories, the performance of random selection slightly decreases despite increased #FLOPs. We hypothesize that this is due to the stochastic nature of long-term

Table 6: Comparison with equalized compute on DCLM 400M-4x, 1B-1x, and 3B-1x settings.

| Method | #FLOPs / #Tokens | Commonsense Reasoning (3 tasks) | Language Understanding (6 tasks) | Reading Comprehension (3 tasks) | Symbolic Problem Solving (5 tasks) | World Knowledge (5 tasks) | Core (22 tasks) |
|---|---|---|---|---|---|---|---|
| **400M-4x Setting:** 412M model | | | | | | | |
| Random | 8.00 ∗1E19 / 32.8B | 0.25335 | 0.28315 | 0.10477 | 0.15643 | 0.22858 | 0.21356 |
| Random | 9.11 ∗1E19 / 37.4B | 0.26988 | **0.28965** | 0.08906 | 0.16319 | **0.23082** | 0.21749 |
| MATES | 9.11 ∗1E19 / 32.8B | 0.28176 | 0.28358 | 0.14225 | 0.16296 | 0.22179 | 0.22260 |
| Group-MATES | 9.11 ∗1E19 / 32.8B | **0.29190** | 0.28735 | **0.14997** | **0.18890** | 0.22908 | **0.23362** |
| **1B-1x Setting:** 1.4B model | | | | | | | |
| Random | 24.00 ∗1E19 / 28.0B | 0.34994 | 0.38584 | 0.22059 | 0.18291 | 0.30784 | 0.29456 |
| Random | 25.71 ∗1E19 / 30.0B | 0.36642 | 0.37954 | 0.22403 | 0.18335 | 0.30665 | 0.29539 |
| MATES | 25.71 ∗1E19 / 28.0B | 0.36331 | 0.39640 | 0.22548 | 0.19958 | 0.30415 | 0.30288 |
| Group-MATES | 25.71 ∗1E19 / 28.0B | **0.36997** | **0.39744** | **0.23922** | **0.20250** | **0.30793** | **0.30747** |
| **3B-1x Setting:** 2.8B model | | | | | | | |
| Random | 9.4 ∗1E20 / 55.9B | 0.44969 | 0.47816 | 0.27832 | 0.18070 | 0.37523 | 0.35603 |
| Random | 9.7 ∗1E20 / 57.7B | 0.45261 | 0.48056 | 0.28435 | 0.18126 | 0.37431 | 0.35782 |
| MATES | 9.7 ∗1E20 / 55.9B | 0.44178 | 0.48263 | 0.30487 | 0.18497 | 0.37799 | 0.36139 |
| Group-MATES | 9.7 ∗1E20 / 55.9B | **0.45874** | **0.48504** | **0.31094** | **0.19591** | **0.38146** | **0.36846** |

Table 7: Results on DCLM 1B-3x setting.

| Method | Commonsense Reasoning (3 tasks) | Language Understanding (6 tasks) | Reading Comprehension (3 tasks) | Symbolic Problem Solving (5 tasks) | World Knowledge (5 tasks) | Core (22 tasks) |
|---|---|---|---|---|---|---|
| **1B-3x Setting:** 1.4B model, 84.0B tokens | | | | | | |
| Random | 0.42433 | 0.44592 | 0.27744 | 0.17986 | 0.35294 | 0.33840 |
| MATES | 0.40872 | **0.45021** | 0.29262 | 0.20697 | 0.34452 | 0.34376 |
| Group-MATES | **0.43300** | 0.44907 | **0.29663** | **0.21737** | **0.36320** | **0.35391** |

LLM pretraining optimization, where simply adding more tokens does not guarantee consistent improvements across all tasks, especially when comparing similar computational budgets [31]. For instance, in the official DCLM leaderboard, 7B-2x models do not universally outperform 7B-1x models despite doubling the training tokens. As highlighted in the DCLM benchmark, Core score provides a more reliable and comprehensive metric that mitigates individual task variability.

## C.3   1B-3x Setup

This experiment illustrates that larger models are more robust to data quality variations and consequently require more extensive training to fully manifest the benefits of data selection. In Table 7, we further run a set of experiments in the 1B-3x setup, where the training tokens are 3 times more than 1B-1x. Our results show that the absolute Core score improvement of Group-MATES over random increases from 1.3% (1B-1x) to 1.6% (1B-3x), doubling the gains achieved by MATES. Therefore, Group-MATES consistently maintains its advantage even as model size and training data scale up.

## C.4   MATES Setup

In this section, we compare Group-MATES with previous pretraining data curation baselines, following the same setup as MATES [50]. These methods include (1) DSIR [47]: proximity to Wikipedia based on n-gram features. (2) SemDeDup [1]: deduplicating semantically similar data. (3) DsDm [11]: static approximation of influence functions by a converged proxy model. (4) QuRating [44]: ranking with educational values distilled from GPT-3.5. As shown in Table 8, Group-MATES achieves the best average downstream results with minimal additional costs, highlighting the potential of optimizing group influences in data-efficient pretraining.

## C.5   Selection Ratio

In Table 9, we explore the impact of varying the selection ratio of Group-MATES to 10%, 25%, and 50%. A 10% selection ratio does not perform as effectively as the other two, likely due to the loss of diversity in a high-quality corpus like DCLM when the selection is too aggressive. Both 25% and 50% achieve comparable results; however, 50% produces more training tokens, making it the preferred choice for our final selection ratio.

Table 8: Zero-shot evaluation of pretraining 1B models with different data selection methods on C4. We report per-task accuracy and the total #FLOPs for each method. All results except Group-MATES are directly copied from the original MATES paper [50]. Best performances are marked **bold**.

| METHOD (#FLOPs *1E19) | SCIQ | ARC-E | ARC-C | LOGIQA | OBQA | BOOLQ | HELLASWAG | PIQA | WINOGRANDE | AVERAGE |
|---|---|---|---|---|---|---|---|---|---|---|
| **1B SETTING:** 1B MODEL, 25B TOKENS | | | | | | | | | | |
| RANDOM (17.67) | 65.8 | 43.7 | 25.6 | 27.5 | 31.8 | 60.2 | 43.8 | 68.9 | 50.7 | 46.4 |
| DSIR (17.67) | 65.8 | 42.6 | 24.7 | 28.7 | 29.2 | 59.7 | 44.2 | 68.3 | **53.2** | 46.3 |
| SEMDEDUP (19.13) | 66.8 | **45.5** | 25.3 | 27.6 | 30.6 | 60.2 | 45.3 | 69.7 | 52.5 | 47.1 |
| DSDM (22.04) | **68.2** | 45.0 | **26.5** | 26.6 | 29.4 | 59.0 | 44.8 | 68.9 | 51.9 | 46.7 |
| QURATING (37.67) | 67.1 | **45.5** | 25.6 | 26.9 | 29.8 | 60.3 | 45.2 | 70.2 | 51.6 | 46.9 |
| MATES (19.97) | 67.3 | 44.9 | 25.9 | 28.7 | 32.2 | **60.9** | 45.3 | 69.5 | 52.4 | 47.5 |
| GROUP-MATES (20.37) | 67.8 | 45.0 | 25.5 | **28.9** | **32.6** | **60.9** | **47.4** | **70.5** | 52.4 | **47.9** |

Table 9: Results on DCLM 400M-4x setting with different selection ratios.

| RATIO | COMMONSENSE REASONING (3 tasks) | LANGUAGE UNDERSTANDING (6 tasks) | READING COMPREHENSION (3 tasks) | SYMBOLIC PROBLEM SOLVING (5 tasks) | WORLD KNOWLEDGE (5 tasks) | CORE (22 tasks) |
|---|---|---|---|---|---|---|
| 10% | 0.27575 | 0.27950 | 0.13437 | **0.19149** | 0.22776 | 0.22743 |
| 25% | 0.28573 | 0.28654 | **0.15217** | 0.18646 | 0.22830 | 0.23212 |
| 50% (OURS) | **0.29190** | **0.28735** | 0.14997 | 0.18890 | **0.22908** | **0.23362** |

## C.6 Design of Relational Data Influence Model

In this section, we vary the design choices of our relational data influence model, including replacing the model backbone with BERT-base [9], choosing dot product or an FFN model as the sim function. As shown in Figure 7a, BERT demonstrates weaker abilities to approximate oracle data influences than BGE, as the latter has been specifically optimized for sentence embeddings. Taking FFN as the sim function does not significantly decrease the approximation performance but introduces additional parameters; choosing dot product, the performance dramatically drops. This validates our choice to align the similarity measurement with the original BGE, i.e., cosine similarity.

## C.7 Number of Collected Trajectories

In this section, we examine the effects of number of collected trajectories on the approximation performance of our relational data influence model. As shown in Figure 7b, scaling up the number of collected trajectory consistently elevates the performance, but with diminishing returns. Considering the effectiveness-efficiency trade-off, we finally choose 20k as the number of collected trajectories.

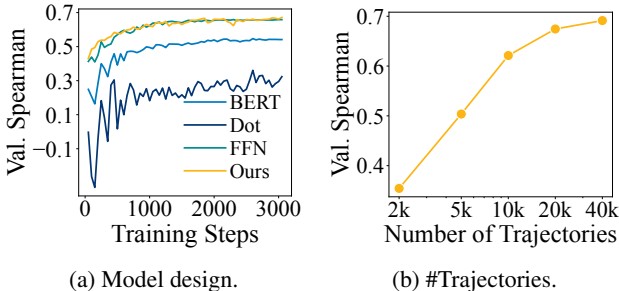

(a) Model design.    (b) #Trajectories.

Figure 7: Performance of relational data influence model with different designs (a) and number of trajectories (b).

## C.8 Rollout Length

In Table 10, we investigate the effect of varying the rollout length $T$ in Group-MATES, considering values of 2, 10, and 20. We observe that $T = 10$ performs slightly better than $T = 2$, suggesting that a moderate increase in rollout length enables our relational data influence model to better capture long-term effects. However, increasing $T$ to 20 results in a performance decline, likely due to the increased complexity of modeling combinatorial effects over longer trajectories, which incurs additional challenges for the relational data influence model. These results indicate the importance

Table 10: Results on DCLM 1B-1x setting with different rollout length $T$.

| $T$ | COMMONSENSE REASONING (3 tasks) | LANGUAGE UNDERSTANDING (6 tasks) | READING COMPREHENSION (3 tasks) | SYMBOLIC PROBLEM SOLVING (5 tasks) | WORLD KNOWLEDGE (5 tasks) | CORE (22 tasks) |
|---|---|---|---|---|---|---|
| 2 | 0.36687 | **0.40461** | 0.22536 | 0.20110 | **0.30817** | 0.30685 |
| 20 | 0.36549 | 0.39901 | 0.21265 | **0.20501** | 0.30081 | 0.30262 |
| 10 (OURS) | **0.36997** | 0.39744 | **0.23922** | 0.20250 | 0.30793 | **0.30747** |

of selecting an appropriate rollout length that sufficiently reflects group-level data influence while remaining tractable for the relational data influence model to learn effectively.

## D Limitations

Our current study focuses on models ranging from 412M to 2.8B parameters, providing initial validation of our proposed methods. However, extending these insights to large-scale, production-level training scenarios remains a promising direction. On one hand, scaling up offers greater flexibility and potential gains for data selection, as the larger candidate pool and increased demand for efficiency make sophisticated curation strategies more valuable, and the relative cost of data selection becomes less significant. On the other hand, large-scale pretraining may introduce new stability and optimization challenges that call for dedicated methodological advances. We leave the exploration of these directions to future work.

Future research could further advance group-level data influence theory itself, for example by characterizing the interactions and dependencies among data groups, analyzing the conditions under which group-level influences significantly diverge from individual data influences, and developing new theoretical frameworks that connect influence modeling with generalization and representation learning. Such work may yield deeper insights into the fundamental principles that govern collective data effects and provide stronger foundations for principled data curation strategies.

## E Broader Impacts

Our work paves the way for a future where efficient pretraining seamlessly integrates data valuation, curation, and model training into a unified, self-optimizing framework. By advancing group-level data selection, our approach empowers foundation models to utilize data wisely and purposefully, significantly reducing computational costs while enhancing scalability and generalization. This breakthrough has the potential to lower resource barriers, making high-performance AI more accessible to a wider range of researchers and organizations.

Beyond efficiency, our work improves the interpretability of training data influence, shedding light on how different subsets contribute to model learning. As foundation models become increasingly capable of dynamically adapting to evolving data distributions, they will drive progress in various fields, from AI-driven scientific discovery to large-scale real-world applications. Moving forward, our approach lays the groundwork for a new paradigm in pretraining—one where models autonomously optimize their learning trajectories with minimal human intervention, leading to more efficient, adaptive, and impactful AI development.

Table 11: Full results on DCLM 400M-4x. The number beside each task denotes the number of few-shot demonstrations used for evaluation. We exclude CommonsenseQA from the core score calculation due to its instability and limited informativeness. For instance, in the original DCLM paper, the 412M model dramatically outperforms the 1.4B model by 76.6% on this task.

| TASK | RANDOM | EDU CLASSIFIER | MATES | QUAD | GROUP-MATES |
|---|---|---|---|---|---|
| AGI_EVAL_LSAT_AR (3) | 0.19565 | **0.28696** | 0.20435 | 0.20000 | 0.27826 |
| ARC_CHALLENGE (10) | 0.29522 | **0.32253** | 0.29863 | 0.29181 | 0.27730 |
| ARC_EASY (10) | 0.57912 | **0.59975** | 0.57323 | 0.58460 | 0.56860 |
| BIGBENCH_CS_ALGORITHMS (10) | **0.44697** | 0.33712 | 0.39697 | 0.43258 | 0.44091 |
| BIGBENCH_DYCK_LANGUAGES (10) | 0.19300 | **0.21600** | 0.18800 | 0.20300 | 0.16500 |
| BIGBENCH_LANGUAGE_IDENTIFICATION (10) | 0.24690 | 0.25320 | 0.25500 | 0.25310 | **0.25750** |
| BIGBENCH_OPERATORS (10) | 0.14762 | 0.18095 | 0.16190 | 0.14762 | **0.20952** |
| BIGBENCH_QA_WIKIDATA (10) | 0.52099 | **0.52492** | 0.52360 | 0.50431 | 0.51557 |
| BIGBENCH_REPEAT_COPY_LOGIC (10) | 0.00000 | 0.03125 | **0.06250** | 0.00000 | 0.03125 |
| BOOLQ (10) | 0.56881 | 0.49021 | 0.59113 | 0.58899 | **0.61407** |
| COMMONSENSE_QA (10) | **0.37838** | 0.22195 | 0.22523 | 0.31286 | 0.20393 |
| COPA (0) | 0.62000 | 0.69000 | 0.66000 | **0.74000** | 0.68000 |
| COQA (0) | 0.21195 | 0.21283 | **0.22836** | 0.21308 | 0.21320 |
| HELLASWAG (10) | 0.45230 | 0.45399 | 0.45519 | 0.45589 | **0.45907** |
| HELLASWAG (0) | 0.45638 | 0.45688 | 0.45828 | 0.45818 | **0.46116** |
| JEOPARDY (10) | 0.12347 | **0.14875** | 0.11854 | 0.09442 | 0.14690 |
| LAMBADA_OPENAI (0) | **0.50708** | 0.45624 | 0.50340 | 0.50010 | 0.50049 |
| MMLU_FEWSHOT (5) | 0.24948 | 0.24992 | 0.22825 | 0.25419 | **0.26629** |
| OPENBOOK_QA | 0.33400 | 0.33600 | **0.34200** | **0.34200** | 0.33400 |
| PIQA (10) | **0.70403** | 0.69369 | 0.70131 | 0.70022 | 0.70185 |
| SQUAD (10) | 0.23709 | 0.23936 | **0.27436** | 0.23094 | 0.25232 |
| WINOGRAD (0) | 0.69231 | **0.70330** | 0.69963 | 0.68864 | 0.69231 |
| WINOGRANDE (0) | 0.54538 | **0.55406** | 0.53354 | 0.52802 | 0.54775 |
| **CORE** | 0.21356 | 0.21821 | 0.22260 | 0.22358 | **0.23362** |

Table 12: Full results on DCLM 1B-1x. The number beside each task denotes the number of few-shot demonstrations used for evaluation. We exclude CommonsenseQA from the core score calculation due to its instability and limited informativeness. For instance, in the original DCLM paper, the 412M model dramatically outperforms the 1.4B model by 76.6% on this task.

| TASK | RANDOM | EDU CLASSIFIER | MATES | QUAD | GROUP-MATES |
|---|---|---|---|---|---|
| AGI_EVAL_LSAT_AR (3) | 0.19565 | 0.23913 | 0.24783 | 0.26522 | **0.27826** |
| ARC_CHALLENGE (10) | 0.36007 | **0.37799** | 0.36092 | 0.34386 | 0.35836 |
| ARC_EASY (10) | 0.65362 | **0.69360** | 0.64689 | 0.64226 | 0.65909 |
| BIGBENCH_CS_ALGORITHMS (10) | 0.44091 | 0.44015 | 0.43485 | **0.44394** | 0.41667 |
| BIGBENCH_DYCK_LANGUAGES (10) | 0.22400 | **0.27300** | 0.23600 | 0.17400 | 0.22600 |
| BIGBENCH_LANGUAGE_IDENTIFICATION (10) | **0.25430** | 0.24940 | 0.24370 | 0.25390 | 0.25410 |
| BIGBENCH_OPERATORS (10) | 0.22381 | 0.22381 | 0.20476 | 0.20000 | **0.23333** |
| BIGBENCH_QA_WIKIDATA (10) | **0.60179** | 0.60066 | 0.59151 | 0.59288 | 0.58531 |
| BIGBENCH_REPEAT_COPY_LOGIC (10) | 0.03125 | 0.06250 | 0.06250 | **0.09375** | 0.06250 |
| BOOLQ (10) | 0.61957 | 0.51315 | 0.61988 | 0.54220 | **0.62538** |
| COMMONSENSE_QA (10) | 0.31368 | 0.21458 | 0.27600 | 0.26536 | **0.33579** |
| COPA (0) | 0.70000 | 0.67000 | **0.72000** | 0.70000 | **0.72000** |
| COQA (0) | 0.30527 | 0.31003 | 0.31204 | **0.31229** | 0.31079 |
| HELLASWAG (10) | 0.57648 | 0.57170 | 0.58156 | 0.57220 | **0.58604** |
| HELLASWAG (0) | 0.58186 | 0.57518 | **0.58335** | 0.57837 | 0.57807 |
| JEOPARDY (10) | 0.24318 | **0.31211** | 0.24653 | 0.26231 | 0.23064 |
| LAMBADA_OPENAI (0) | 0.59441 | 0.55055 | 0.60120 | 0.59402 | **0.60489** |
| MMLU_FEWSHOT (5) | 0.25699 | 0.25345 | 0.25423 | 0.25644 | **0.27533** |
| OPENBOOK_QA | 0.38400 | **0.39200** | 0.38000 | 0.37000 | 0.38600 |
| PIQA (10) | 0.73558 | 0.74102 | 0.73830 | **0.74483** | 0.74429 |
| SQUAD (10) | 0.35762 | **0.41183** | 0.36471 | 0.39773 | 0.39272 |
| WINOGRAD (0) | 0.74359 | 0.75458 | 0.78755 | **0.79853** | 0.77656 |
| WINOGRANDE (0) | 0.58800 | 0.58011 | 0.57380 | 0.57853 | **0.59116** |
| **CORE** | 0.29456 | 0.29257 | 0.30288 | 0.29340 | **0.30747** |

