# OpenReview forum: "Group-Level Data Selection for Efficient Pretraining"
_NeurIPS.cc/2025/Conference — NeurIPS 2025 poster_

### Official Review · Reviewer_NRhp · 2025-07-01

**Clarity:** 2
**Significance:** 2
**Originality:** 2
**Rating:** 4
**Confidence:** 1

**Summary:**

This paper introduces a new data selection approach Group-MATES to optimize the speed-quality frontier of LLM pretraining. The relational data influence model is introduced and is used as a heuristic to approximate data influence. Experiments show improvement over SOTA.

**Questions:**

1. What are the parameters to be optimized in \Theta_rel? What is w_o? I see some explaination for equation (7) but that's for MATES.
2. For Appendix A.2 Connection between Relationship Weight and Trajectory-Specific Influence Function, "relationship weight R in Eq. 14 serves a similar purpose by reweighting the influence of a data point based on its relationships with other points in the training trajectory" this seems quite a large jump to the heuristics you are using. In Section 3, the discussions are on individual data influences (group-level influences are also approximated by decomposition into individual influences). Can you provide more insights on why you use heuristics that consider interactive effects?
3. What is the computational overhead of Group-MATES? It seems the computation of (13) grows linearly with t as it considers the interaction of x_t with all previous data?
4. How do Group-MATES compare with MATES?

**Ethical Concerns:**

["NO or VERY MINOR ethics concerns only"]

**Final Justification:**

In the responses, authors addressed my concerns in great detail. Therefore, I raise my score.
However, since this is not in my area, I am not confident about my review.

**Limitations:**

yes

**Quality:**

2

**Strengths And Weaknesses:**

Strengths: The authors have carried out extensive experiments and the experimental setup are detailed and well explained.

Weakness: The algorithm description is unclear and hard to follow. Despite the presence of Figure 2, I still struggle to understand how algorithm works. Notations in 4.1 are introduced without proper explanation (13) - (15).
"Theoretical analysis" in A.2 seems forced and not directly related to (13) - (15). The intuition is not clear from the explanation. (See questions below)

---

> ### Author Rebuttal · Authors · 2025-07-30
>
> Thank you for your time and insightful review of our paper! We address your questions/comments below:
>
> **Question 1:** What are the parameters to be optimized in \Theta_rel? What is w_o?
>
> **Response:** In Section 4, we reuse the notation from our preliminary section (L133-L136). Therefore, the trainable parameters of $\Theta_{\text{rel}}$ still consist of two parts, (1) a pretrained base language model that produces sentence representation $\textbf{h}\_{x_i}$ and (2) a regression head $\textbf{w}_o$ with input_dim = hidden size and output_dim = 1. The initialization of these two parts is described in L237-L238, with further details provided in L502-L511. We will reiterate the definitions of these notations in Section 4 in the revised version.
>
> ---
>
> **Question 2:** In Section 3, the discussions are on individual data influences (group-level influences are also approximated by decomposition into individual influences). Can you provide more insights on why you use heuristics that consider interactive effects?
>
> **Response:** As revealed by prior studies [1, 2], group-level influences often deviate from the sum of individual influences since the interactions among data points can sometimes cancel out or amplify individual influences. To capture these interactive dynamics, we build our relational data influence model with relationship weights (Eq. 14) to capture the interactive effects among training data. A case study illustrating such effects is presented in Appendix C.7, and its outcome aligns with intuitive expectations. We will add a reference to this study in the main text.
>
> We mention trajectory-specific influence function in Appendix A.2, as we find it an interesting theoretical tool that quantifies data influence given the training trajectory. However, computing exact Hessian products for each training data (i.e., Eq. 40) is significantly more expensive than our method that parameterizes data influence with a relational model. Therefore, in their original paper [3], the primary setup is pretraining on 1% of the Pile dataset (~3B tokens), while ours can efficiently scale to 56B tokens (3B-1x) and beyond. We are willing to incorporate direct performance comparisons with this recent work once it is open-sourced.
>
> ---
>
> **Question 3:** What is the computational overhead of Group-MATES? The computation of (13) grows linearly with t as it considers the interaction of x_t with all previous data?
>
> **Response:** We have shown the computational overhead measured by FLOPs in Appendix C.1 and by wall clock time in the table below. The findings are consistent: Group-MATES accounts for only a very small portion (e.g., 3.4% in 3B-1x setup) of the total computation, and the relative overhead is generally smaller in larger setups since pretraining FLOPs dominate the computation. We will move this computational analysis to the main text.
>
> |                                               | Group-MATES (400M-4x) | MATES (400M-4x) | Brute-Force Group Selection (400M-4x) | Group-MATES (1B-1x) | MATES (1B-1x) | Group-MATES (3B-1x) | MATES (3B-1x) |
> | --------------------------------------------- | --------------------- | --------------- | ------------------------------------- | ------------------- | ------------- | ------------------- | ------------- |
> | Model Pretraining (H100 Hours)                | 104                   | 104             | 104                                   | 240                 | 240           | 740                 | 740           |
> | Oracle Data Influence Collection (H100 Hours) | 4.5                   | 1.8             | 22632240                              | 12.1                | 4.8           | 18.7                | 7.5           |
> | Data Influence Model Training (H100 Hours)    | 2.7                   | 2.0             | -                                     | 2.7                 | 2.0           | 2.7                 | 2.0           |
> | Data Influence Model Inference (H100 Hours)   | 14.2                  | 14.2            | -                                     | 12.0                | 12.0          | 23.9                | 23.9          |
> | Data Selection (208-Core CPU Time)            | 10min                 | 1min            | 151min                                | 8min                | 1min          | 17min               | 2min          |
>
> The computation of Eq. 13 indeed grows linearly with $t$ (or our selected size $n$), which motivates our design of cluster-based selection (Section 4.3). By distributing the selected size into separate clusters, we significantly reduce the time complexity by $d$ (the number of clusters). Furthermore, our data selection can be performed solely on CPUs after one-time data influence model inference (to get $\textbf{h}\_{x_i}$ and $\textbf{w}_o\cdot\textbf{h}\_{x_i}$), and is highly parallelizable—e.g., different clusters can utilize separate cores for independent selection. The actual efficiency gain of this cluster-based method is demonstrated in Figure 6a.
>
> ---
>
> **Question 4:** How do Group-MATES compare with MATES?
>
> **Response:** From the methodology perspective, MATES approaches individual data selection by modeling individual data influences given a certain model state (Eq. 7), whereas Group-MATES approaches group-level data selection by modeling both individual data influences as well as the relationships among data points given model training trajectories (Eq. 13). As a language model is ultimately trained on a group of data step-by-step, the selection procedure of Group-MATES better aligns with the actual training process.
>
> To fairly compare Group-MATES with MATES, we report **Core score/total FLOPs (1e19)/total GPU hours** in the table below. “Total” here means we consider all costs from data selection and model pretraining. We can see that Group-MATES only incurs **~3% additional FLOPs/wall clock time** compared to MATES, yet **doubles the net efficiency gain** (compute reduction for reaching a certain performance) of MATES over random selection across different scales. This highlights the benefits of pursuing group-level selection with our relational data influence model.
>
> *400M-4x (cells used to calculate net efficiency gain are marked italic):*
>
> | Tokens (B)           | Random           | Quad           | MATES           | Group-MATES     |
> | -------------------- | ---------------- | -------------- | --------------- | --------------- |
> | 19.7                 | 0.193/4.8/62     | 0.196/5.8/85   | 0.196/5.1/73    | 0.198/5.2/75    |
> | 26.2                 | 0.208/6.4/83     | 0.216/8.4/114  | 0.214/7.0/98    | *0.222/7.1/100* (net efficiency gain = 31.5%) |
> | 32.8                 | 0.214/8.0/104    | 0.224/11.0/142 | *0.223/8.9/122* (net efficiency gain = 16.4%) | 0.234/9.1/125   |
> | 39.4 (for reference) | 0.218/9.6/125    | -              | -               | -               |
> | 45.9 (for reference) | *0.221/11.2/146* | -              | -               | -               |
>
> *1B-1x (cells used to calculate net efficiency gain are marked italic):*
>
> | Tokens (B)           | Random           | Quad           | MATES            | Group-MATES      |
> | -------------------- | ---------------- | -------------- | ---------------- | ---------------- |
> | 16.8                 | 0.257/14.4/144   | 0.260/15.3/159 | 0.262/14.8/155   | 0.266/15.0/160   |
> | 22.4                 | 0.283/19.2/192   | 0.288/21.0/212 | 0.291/19.9/207   | 0.298/20.3/213   |
> | 28.0                 | 0.295/24.0/240   | 0.293/26.8/265 | *0.303/25.1/259* (net efficiency gain = 10.1%) | *0.308/25.7/267* (net efficiency gain = 20.5%) |
> | 33.6 (for reference) | *0.304/28.8/288* | -              | -                | -                |
> | 39.2 (for reference) | *0.308/33.6/336* | -              | -                | -                |
>
> ---
>
> We sincerely thank you again for your valuable time and thoughtful review. We're delighted that you find our experiments extensive and well explained. In our revised version, we will clarify notations in our method section, add an overview algorithm for our method, and highlight the computational overhead analysis in the main text.
>
> [1] Hu, Yuzheng, et al. "Most Influential Subset Selection: Challenges, Promises, and Beyond." NeurIPS 2024.
>
> [2] Huang, Jenny Y., et al. "Approximations to worst-case data dropping: unmasking failure modes." arXiv preprint arXiv:2408.09008 (2024).
>
> [3] Wang, Jiachen T., et al. "Capturing the Temporal Dependence of Training Data Influence." ICLR 2025.

---

> ### Author Response · Authors · 2025-08-06
>
> Dear Reviewer,
>
> Thank you once again for taking the time to review our paper. If there are any additional concerns or questions, please do not hesitate to let us know. We would be happy to address them promptly.
>
> Thank you!

---

> ### Author Response · Authors · 2025-08-08
> **Further explanation of our method pipeline**
>
> Dear Reviewer NRhp,
>
> As the discussion period is about to end, we would like to further explain our method pipeline to address your concerns of clarity.
>
> 1) **Collect oracle influences with rollouts**
>
> - Start from the current pretrained model $\mathcal{M}$.
> - Use a **random rollout policy** $\pi_{\text{rand}}$, which at each step $t$ uniformly samples the next data point $x_t$, update the model $\mathcal{M}_{t+1} = \mathcal{A}(\mathcal{M}_t, x_t)$, and record its oracle influence $\mathcal{I}(\mathcal{M}_t, x_t) = \mathcal{L}(\mathcal{D}_r \mid \mathcal{M}\_{t+1}) - \mathcal{L}(\mathcal{D}_r \mid \mathcal{M}_t)$. $\mathcal{D}\_{(t-1)} = \set{{x_i}}\_{i<t}$.
> - The collected $(x_t, \mathcal{D}_{(t-1)}, \mathcal{I}(\mathcal{M}_t, x_t))$ triplets from these random rollouts form the **initial supervision set** for training the relational model below.
>
> 2) **Train and bootstrap the relational data influence model**
>
> - Train $\Theta^{\text{rel}}_{\text{init}}$ on the initial supervision set to predict $\mathcal{I}(\mathcal{M}\_t, x_t)$ from the candidate’s embedding $\textbf{h}\_{x_t}$ and relationship weights $R\_{x_i,x_t} = \text{sim}(\textbf{h}\_{x_i}, \textbf{h}\_{x_t})$ with previous embeddings $\set{\textbf{h}\_{x_i}}\_{i<t}$ using the parametric form in Eq. 13.
> - Define a **bootstrapping rollout policy** $\pi_{\text{boot}}$ that, at each step $t$, samples from the lowest and highest predicted influences under $\Theta^{\text{rel}}_{\text{init}}$.
> - Generate additional trajectories with $\pi_{\text{boot}}$ and combine them with the random rollout trajectories to form an **expanded supervision set**.
> - Retrain the model on this combined set to obtain $\Theta^{\text{rel}}_{\text{final}}$.
>
> 3) **Cluster-based inference for group selection**
>
> - Partition the pool $\mathcal{D}$ into $d$ clusters $\{\mathcal{C}^1, \ldots, \mathcal{C}^d\}$ using influence-aware clustering with relationship weight $R$ as the similarity metric.
> - In each cluster $\mathcal{C}^i$, run the sequential selection with $\Theta^{\text{rel}}_{\text{final}}$, allocating $\left\lceil n \cdot \frac{|\mathcal{C}^i|}{|\mathcal{D}|} \right\rceil$ selection budget and iteratively choosing $\arg\min\_{x_j \in \mathcal{C}^i \setminus \mathcal{C}^i\_{(t-1)}} \Theta^{\text{rel}}\_{\text{final}}(x_j \mid \mathcal{C}^i\_{(t-1)})$.
> - Merge selections from all clusters to obtain the final subset $\mathcal{D}_{(n)}$.
> - Continue pretraining the model $\mathcal{M}$ on $\mathcal{D}_{(n)}$ until the next selection round, then repeat the process starting from Step 1. More details can be found in our response to Reviewer zK2t Q1.
>
> In our original writing flow, we began with the formulation of our relational data influence model to better connect our method with the group-level selection described in Section 3. In the next version, we will refine the writing to present our method more concisely and clearly. Thank you again for your time and effort in reviewing our paper!

---

> > ### Comment · Reviewer_NRhp · 2025-08-08
> >
> > Thanks for the detailed response. I think you answered my question well and I will raise my score.

---

> > > ### Author Response · Authors · 2025-08-08
> > >
> > > Thank you very much for recognizing our rebuttal!
> > >
> > > We will carefully revise the paper based on your feedback and incorporate all of the results mentioned above.

---

### Official Review · Reviewer_aQV7 · 2025-07-02

**Clarity:** 2
**Significance:** 2
**Originality:** 3
**Rating:** 4
**Confidence:** 3

**Summary:**

This paper studies the data selection problem in the pre-training stage, which is an important issue. The paper differs itself from previous works in that it studies the combinatorial effect of different data group. To avoid tedious effort, the group-level selection is parameterized by a relational data influence model. The evaluation is conducted on DCLM at different scales and shows sufficient improvement.

**Questions:**

1. It appears that the improvement brought by Group-MATES reduces as the LLM model size increases. Does this mean that eventually all data selection will degrade to random selection, considering the size of SOTA LLMs?
2. What are the differences between data selection in classic ML settings and LM settings?

**Ethical Concerns:**

["NO or VERY MINOR ethics concerns only"]

**Final Justification:**

Final Justification

**Limitations:**

Yes

**Quality:**

3

**Strengths And Weaknesses:**

Pros:

1. The paper is generally well-written. The majority of its sections are relatively easy to read and understand.
2. The intuition of the proposed method is clear and the proposed method, especially the parameterization of group-level selection, is backed by theoretical analysis.
3. The proposed method demonstrates strong empirical results under different evaluation settings, showcasing the superiority of this approach.

Cons:

1. Figure 1a is not very understandable, which may harm the readability of the introduction. The same goes for Fig 6b.
2. The evaluation scope seems limited, only on the DCLM benchmark. Although the related LM benchmark is limited, extending the methods to other NLP tasks could better prove the consistent improvement under different settings.
3. The content of the paper is heavily chunked due to the page limit. Please consider re-organizing the sections for better readability.
4. Some of the content lacks further discussion, especially the content in Appendix C.

---

> ### Author Rebuttal · Authors · 2025-07-30
>
> Thank you for your time and insightful review of our paper! We address your questions/comments below:
>
> **Weakness 1:** Figure 1a/6b is not very understandable, which may harm the readability of the introduction.
>
> **Response:** Thank you for your suggestions! Figure 1a serves as our motivation that individual data selection (Eq. 8) often deviates from brute-force group selection (Eq. 9-11) as the dataset size increases. Details about the experimental setup and the definition of overlap are provided in our preliminary section (L141-L155). We will move Figure 1 directly into Section 3 for better readability.
>
> In Figure 6b, we analyze the distribution of relationship weights $R$ (Eq. 14) between data points within the same cluster (intra-cluster) or across different clusters (inter-cluster). To enable efficient inference, we compute relationship weights only within each cluster in the selection process (L197-L198), and Figure 6b illustrates that this design preserves essential relationship information, as inter-cluster relationship weights are distributed around 0. We will add this clarification to the paper.
>
> ---
>
> **Weakness 2:** The evaluation scope seems limited, only on the DCLM benchmark. Extending the methods to other NLP tasks could better prove the consistent improvement under different settings.
>
> **Response:** The benchmark used in our main experiments, DCLM, is a standardized pretraining data curation benchmark that unifies the data selection pool, training scripts, and **22 Core evaluation tasks** that holistically and robustly evaluate the pretrained models (L211-L225). It has been widely adopted as the primary setup in a line of recent works on data selection [1], data mixing [2], and data synthesis [3], reflecting its status as a commonly recognized and authoritative benchmark for pretraining data curation.
>
> To demonstrate the generalization ability of our method, we also conduct experiments on the C4 dataset in Appendix C.6, following the identical setup in MATES to select 25B tokens from a 125B-token pool. As shown in Table 7, Group-MATES consistently matches or surpasses the performance of the previous best method, MATES, on **8 out of 9** evaluation tasks. We will add a pointer to this setup in the main text to better guide readers.
>
> ---
>
> **Weakness 3 & 4:** The content of the paper is heavily chunked due to the page limit. Please consider re-organizing the sections for better readability. Some of the content lacks further discussion, especially the content in Appendix C.
>
> **Response:** Thank you for your suggestions! Appendix C demonstrates the additional results that are not able to be covered in the main content due to the page limit. However, we will re-organize these based on your comments. Specifically, we will:
>
> 1. Move “C.1 FLOPs breakdown” and “C.2 Equalization for Compute” to main text to highlight the low additional cost and net efficiency gain of Group-MATES.
> 2. Form “C.3 Selection Ratio”, “C.4 Design of Relational Data Influence Model”, and “C.5 Number of Collected Trajectories” into one section, as they are all ablations to validate our design choices and hyperparameters.
> 3. Make C.6 “Comparison in MATES Setup” a separate section, as it is a new setup to show the generalization ability of our method.
> 4. Add more cases to facilitate insightful discussions in C.7 Case Study.
>
> ---
>
> **Question 1:** The improvement brought by Group-MATES reduces as the LLM model size increases. Does this mean that eventually all data selection will degrade to random selection, considering the size of SOTA LLMs?
>
> **Response:** This is an interesting question. It is a common finding [4] that larger models are more robust to data quality variations and consequently require more extensive training to fully manifest the benefits of data selection. To validate this, we further run a set of experiments on 1B-3x setup, where the training tokens are 3 times more than 1B-1x setup. Results are shown below:
>
> | 1B-3x       | Commonsense Reasoning | Language Understanding | Reading Comprehension | Symbolic Problem | World Knowledge | Core        |
> | :---------- | :-------------------- | :--------------------- | :-------------------- | :--------------- | :-------------- | :---------- |
> | Random      | 0.42433               | 0.44592                | 0.27744               | 0.17986          | 0.35294         | 0.33840     |
> | MATES       | 0.40872               | **0.45021**            | 0.29262               | 0.20697          | 0.34452         | 0.34376     |
> | Group-MATES | **0.43300**           | 0.44907                | **0.29663**           | **0.21737**      | **0.36320**     | **0.35391** |
>
> We can see that the absolute Core score improvement of Group-MATES over random increases from 1.3% (1B-1x) to 1.6% (1B-3x), doubling the gains achieved by MATES. These results demonstrate that Group-MATES consistently maintains its performance advantage even as model size and training data scale up. We will add it in the paper.
>
> Data selection is a very common and essential practice in building state-of-the-art LLMs, as highlighted in Gemini [5], Qwen [6], and DeepSeek [7] reports. In general, web-crawled data is often noisy and necessitates extensive filtering and delicate algorithms to select high-quality data. Our work proposes a novel group-level selection framework that maximizes group-level data utility at a low cost, and we believe it remains effective in production-level pretraining data construction, as group-level selection is still an underexplored yet critical topic.
>
> ---
>
> **Question 2:** What are the differences between data selection in classic ML settings and LM settings?
>
> **Response:** We summarize three key points that characterize the differences between data selection in classic machine learning and language model pretraining:
>
> 1. **Data Scale:** The scale of data used in LLM pretraining is often orders of magnitude larger than that in traditional ML settings. This significant difference poses challenges for many conventional methods (e.g., Shapley value [8]), which are computationally tractable in classic computer vision tasks such as CIFAR-10 and CIFAR-100, but become infeasible for large-scale pretraining data selection.
> 2. **Data Quality:** Classic ML settings typically operate on supervised datasets where most data is already of high quality. In contrast, the raw web corpus in LM pretraining is often noisy, providing greater potential gains from identifying and selecting high-quality subsets of data.
> 3. **Complexity:** LM data selection is not a single-step process, but a combinational and multi-stage pipeline that includes rule-based filtering, deduplication, and quality assessment. Each stage provides its distinct value. Our work focuses on the later stages of this pipeline, where basic cleaning has been performed and model-oriented selection yields more substantial benefits.
>
> ---
>
> We sincerely thank you again for your valuable time and thoughtful review. We're delighted that you find our paper well-written, recognize the motivation of our method, and highlight our superior performance. In our revised version, we will re-organize the paper for better readability and add 1B-3x results.
>
> [1] Zhang, Jifan, et al. "GPT-4o as the Gold Standard: A Scalable and General Purpose Approach to Filter Language Model Pretraining Data." arXiv preprint arXiv:2410.02755 (2024).
>
> [2] Wettig, Alexander, et al. "Organize the Web: Constructing Domains Enhances Pre-Training Data Curation." ICML 2025.
>
> [3] Nguyen, Thao, et al. "Recycling the Web: A Method to Enhance Pre-training Data Quality and Quantity for Language Models." arXiv preprint arXiv:2506.04689 (2025).
>
> [4] Chang, Ernie, et al. "Scaling Parameter-Constrained Language Models with Quality Data." EMNLP 2024.
>
> [5] Comanici, Gheorghe, et al. "Gemini 2.5: Pushing the frontier with advanced reasoning, multimodality, long context, and next generation agentic capabilities." arXiv preprint arXiv:2507.06261 (2025).
>
> [6] Yang, An, et al. "Qwen3 technical report." arXiv preprint arXiv:2505.09388 (2025).
>
> [7] Liu, Aixin, et al. "Deepseek-v2: A strong, economical, and efficient mixture-of-experts language model." arXiv preprint arXiv:2405.04434 (2024).
>
> [8] Cai, Huaiguang. "CHG Shapley: Efficient Data Valuation and Selection towards Trustworthy Machine Learning." arXiv preprint arXiv:2406.11730 (2024).

---

> ### Author Response · Authors · 2025-08-06
>
> Dear Reviewer,
>
> Thank you once again for taking the time to review our paper. If there are any additional concerns or questions, please do not hesitate to let us know. We would be happy to address them promptly.
>
> Thank you!

---

> ### Comment · Reviewer_aQV7 · 2025-08-08
>
> Thanks to the authors for their detailed rebuttal. I think the rebuttal solves my concern. I will keep my score as positive. However, I will not raise my score to 5, mainly given that parts of the paper are hard to understand. Thanks!

---

> > ### Author Response · Authors · 2025-08-09
> >
> > Thank you very much for recognizing our rebuttal!
> >
> > We will carefully revise the paper to improve its clarity and incorporate all of the results presented above.

---

### Official Review · Reviewer_aEny · 2025-07-02

**Clarity:** 2
**Significance:** 2
**Originality:** 2
**Rating:** 4
**Confidence:** 1

**Summary:**

The paper proposed Group-MATES as a variant of the MATES method using clustered groups on relational data influence scores and selecting in a parallel way, which enables efficient data selection.

**Questions:**

1. Have you compared the exact wall clock time of MATES and Group-MATES, including the breakdown of data selection process and the training process?
2. Why is Figure 3 only containing the random baseline? Is there results from other baselines available?

**Ethical Concerns:**

["NO or VERY MINOR ethics concerns only"]

**Final Justification:**

The rebuttal solved most concerns but I am not an expert in the domain so I raise my score but with low confidence.

**Limitations:**

Yes

**Quality:**

2

**Strengths And Weaknesses:**

Strengths:
1. The paper is well-written and clearly presented.
2. The proposed method has a significant improvement ovoer random baselines.
3. Comprehensive ablation studies have been provided.

Weaknesses:
1. While the speed-quality frontier is studied, it is using the number of tokens/flops. But the parallel nature of group selection should make is more efficient in wall-clock time but there seems to be no discussion or result on that. See Q1.
2. Figure 3 seems misleading since it only compares with the random baseline. The performance improvement over other baselines seems rather marginal in Table 1, while I imagine the result would be more significant if the proposed method excels at wall-clock time. See Q2.
3. The experiment is only conducted on one dataset/benchmark which might be limited, but I am not an expert in this domain so correct me if I am wrong.

---

> ### Author Rebuttal · Authors · 2025-07-30
>
> Thank you for your time and insightful review of our paper! We address your questions/comments below:
>
> **Weakness 1 + Question 1:** Have you compared the exact wall clock time of MATES and Group-MATES, including the breakdown of data selection process and the training process?
>
> **Response:** In the table below, we demonstrate the actual wall clock time for each part of our method, compared with MATES and brute-force selection. The main difference between Group-MATES and MATES lies in the oracle data influence collection step, where Group-MATES collects 20k rollout trajectories (each containing 10 oracle data influences) and MATES collects 80k locally probed oracle data influences. However, this gap only accounts for **less than 3%** of the total wall clock time relative to pretraining hours. Notably, our data selection can be performed solely on CPUs after the data influence model inference, and is highly parallelizable—e.g., different clusters can utilize separate cores for independent selection. Considering the 2x net efficiency gain achieved by Group-MATES compared to MATES (see next response), this additional wall clock time is negligible.
>
> |                                               | Group-MATES (400M-4x) | MATES (400M-4x) | Brute-Force Group Selection (400M-4x) | Group-MATES (1B-1x) | MATES (1B-1x) | Group-MATES (3B-1x) | MATES (3B-1x) |
> | --------------------------------------------- | --------------------- | --------------- | ------------------------------------- | ------------------- | ------------- | ------------------- | ------------- |
> | Model Pretraining (H100 Hours)                | 104                   | 104             | 104                                   | 240                 | 240           | 740                 | 740           |
> | Oracle Data Influence Collection (H100 Hours) | 4.5                   | 1.8             | 22632240                              | 12.1                | 4.8           | 18.7                | 7.5           |
> | Data Influence Model Training (H100 Hours)    | 2.7                   | 2.0             | -                                     | 2.7                 | 2.0           | 2.7                 | 2.0           |
> | Data Influence Model Inference (H100 Hours)   | 14.2                  | 14.2            | -                                     | 12.0                | 12.0          | 23.9                | 23.9          |
> | Data Selection (208-Core CPU Time)            | 10min                 | 1min            | 151min                                | 8min                | 1min          | 17min               | 2min          |
>
> ---
>
> **Weakness 2 + Question 2:** Why is Figure 3 only containing the random baseline? Are there results from other baselines available?
>
> **Response:** Yes, all baseline results are available except for WebOrganizer, for which we directly report the final results from their paper, as their code and data are not fully open-sourced yet. Figure 3 intends to show the net efficiency gain of Group-MATES over random. The table below shows the comprehensive result. For each method, we report **Core score/total FLOPs (1e19)/total GPU hours** in each cell. “Total” here means we consider all costs from data selection and model pretraining.
>
> *400M-4x (cells used to calculate net efficiency gain are marked italic):*
>
> | Tokens (B)           | Random           | Quad           | MATES           | Group-MATES     |
> | -------------------- | ---------------- | -------------- | --------------- | --------------- |
> | 19.7                 | 0.193/4.8/62     | 0.196/5.8/85   | 0.196/5.1/73    | 0.198/5.2/75    |
> | 26.2                 | 0.208/6.4/83     | 0.216/8.4/114  | 0.214/7.0/98    | *0.222/7.1/100* |
> | 32.8                 | 0.214/8.0/104    | 0.224/11.0/142 | *0.223/8.9/122* | 0.234/9.1/125   |
> | 39.4 (for reference) | 0.218/9.6/125    | -              | -               | -               |
> | 45.9 (for reference) | *0.221/11.2/146* | -              | -               | -               |
>
> *1B-1x (cells used to calculate net efficiency gain are marked italic):*
>
> | Tokens (B)           | Random           | Quad           | MATES            | Group-MATES      |
> | -------------------- | ---------------- | -------------- | ---------------- | ---------------- |
> | 16.8                 | 0.257/14.4/144   | 0.260/15.3/159 | 0.262/14.8/155   | 0.266/15.0/160   |
> | 22.4                 | 0.283/19.2/192   | 0.288/21.0/212 | 0.291/19.9/207   | 0.298/20.3/213   |
> | 28.0                 | 0.295/24.0/240   | 0.293/26.8/265 | *0.303/25.1/259* | *0.308/25.7/267* |
> | 33.6 (for reference) | *0.304/28.8/288* | -              | -                | -                |
> | 39.2 (for reference) | *0.308/33.6/336* | -              | -                | -                |
>
> We can see that Group-MATES achieves **31.5%** and **20.5%** net efficiency gains (measured by GPU hours) in 400M-4x and 1B-1x setups, respectively. By contrast, MATES only achieves **16.4%** and **10.1%** net efficiency gains in 400M-4x and 1B-1x setups. Therefore, Group-MATES nearly **doubles** the net efficiency gain of MATES, further validating the advantage of our group-level selection over individual selection. We will include all baseline comparisons in Figure 3 and add a plot with GPU hours as the x-axis.
>
> ---
>
> **Weakness 3:** The experiment is only conducted on one dataset/benchmark which might be limited.
>
> **Response:** The benchmark used in our main experiments, DCLM, is a standardized pretraining data curation benchmark that unifies the data selection pool, training scripts, and **22 Core evaluation tasks** that holistically and robustly evaluate the pretrained models (L211-L225). It has been widely adopted as the primary setup in a line of recent works on data selection [1], data mixing [2], and data synthesis [3], reflecting its status as a commonly recognized and authoritative benchmark for pretraining data curation.
>
> To demonstrate the generalization ability of our method, we also conduct experiments on the C4 dataset in Appendix C.6, following the identical setup in MATES to select 25B tokens from a 125B-token pool. As shown in Table 7, Group-MATES consistently matches or surpasses the performance of the previous best method, MATES, on **8 out of 9** evaluation tasks. We will add a pointer to this setup in the main text to better guide readers.
>
> ---
>
> We sincerely thank you again for your valuable time and thoughtful review. We're delighted that you find our paper well-written, recognize the effectiveness of our method, and highlight our comprehensive ablation studies. In our revised version, we will compare different methods w.r.t. wall clock time and put all baselines in Figure 3.
>
> [1] Zhang, Jifan, et al. "GPT-4o as the Gold Standard: A Scalable and General Purpose Approach to Filter Language Model Pretraining Data." arXiv preprint arXiv:2410.02755 (2024).
>
> [2] Wettig, Alexander, et al. "Organize the Web: Constructing Domains Enhances Pre-Training Data Curation." ICML 2025.
>
> [3] Nguyen, Thao, et al. "Recycling the Web: A Method to Enhance Pre-training Data Quality and Quantity for Language Models." arXiv preprint arXiv:2506.04689 (2025).

---

> > ### Comment · Reviewer_aEny · 2025-08-07
> >
> > Thank you for the rebuttal. It mostly solved the concern and I will raise my score.

---

> > > ### Author Response · Authors · 2025-08-07
> > >
> > > Thank you very much for recognizing our rebuttal!
> > >
> > > We will carefully revise the paper based on your feedback and incorporate all of the results mentioned above.

---

> ### Author Response · Authors · 2025-08-06
>
> Dear Reviewer,
>
> Thank you once again for taking the time to review our paper. If there are any additional concerns or questions, please do not hesitate to let us know. We would be happy to address them promptly.
>
> Thank you!

---

### Official Review · Reviewer_zK2t · 2025-07-08

**Clarity:** 3
**Significance:** 2
**Originality:** 3
**Rating:** 5
**Confidence:** 3

**Summary:**

This paper focuses on trying to improve pretraining data selection by considering the utility of groups of training examples rather than just considering the influence of a training example individually. As a proof of concept, the group data selection is first done in a computationally expensive, brute force manner. The authors then propose a computationally cheaper approximation called Group-MATES for doing group-level data selection. This method is then tested on training on setups from the DCLM competition at 3 different scales. Finally ablations are performed on the various components of Group-MATES.

**Questions:**

1. The paper says the results in Table 1 are for models pretrained from scratch. At what step in training do you start your data rollouts for training the relational data influence model?

2. Did you try different similarity functions or embedding models for the relational data influence model?

**Ethical Concerns:**

["NO or VERY MINOR ethics concerns only"]

**Final Justification:**

The gains from Group-MATES over MATES are modest which is still reflected in my significance score but the revisions from the rebuttal now give the reader a much clearer picture of the computational cost and relative efficiency gains of these methods. I thus recommend accept provided the revisions from the rebuttal are included in the final draft of the paper (wall clock time measurements and the MATES baseline is included in Figure 3) .

**Limitations:**

Yes, the authors have adequately discussed the limitations.

**Quality:**

4

**Strengths And Weaknesses:**

**Strengths**

Section 2 and 3 clearly lays out previous work and the motivation for the paper. These introductory sections end with an experiment to demonstrate "an empirical gap between group and individual data influence oracles." Starting with an intermediate checkpoint in the DCLM 400M-4x baseline, the goal is to choose $n$ data points that will best minimize the loss in the subsequent training steps. As a baseline, $n$ data points are chosen based on their individual oracle influences. For group-level influences, optimizing directly over the groups results in too large of a search space so instead the authors use a sequential process where the next data point is chosen greedily. The single point with minimal oracle individual influence is added to the selected data and a gradient step is taken with this point; the process is then repeated at each training step with the already chosen data points excluded. Figure 1 shows this results in very different data selection than the individual selection baseline and a substantial improvement in performance on downstream tasks.

The problem is this brute-force group selection is prohibitively expensive, as the oracle individual influences must be computed at each step. The paper proposes Group-MATES as a cheaper approximation. It builds on MATES which approximates individual data influence with a parametric model (a learned regression model on the activations of the last hidden layer of the language model). This is then scaled by a term dependent on the sum of some similarity metric between the data point under consideration and those already selected (i.e. the individual data influence is weighted according to its relation to the selected set) with trainable parameters. The parameters of the relational data influence model are trained by sampling data trajectories, computing the oracle individual influence for the selected data, and then minimizing the mean squared error between this and the proposed model. In short, the goal is for the trained model to approximate the oracle individual influences at each step of training for any individual data point so that this does not have to be computed as in the brute-force method. A final approximation and computational savings is provided by doing the procedure on clusters of the data rather than as a whole.

The method is compared to MATES and random selection in Table 1 and outperforms both these methods.



**Weaknesses**

While Group-MATES does outperform MATES, the gains are modest compared to this baseline and there remains a large gap between Group-MATES and the brute-force group selection (see Figure 4). Thus while the paper does make progress towards approximating this brute-force group selection, it is unclear whether the additional computational work (and overhead to set up the method) is worthwhile for practitioners over simply using MATES. In addition, all the computational comparisons (e.g. Table 5) are done in terms of FLOPS but I would hypothesize a number of the steps for Group-MATES are less efficient in terms of throughput than training steps (i.e. equal FLOPS would not lead to equal wall clock time). However, I would appreciate clarification from the authors on this based on their experiments. Overall, these observations are primarily reflected in my significance score.

I think the right panel of Fig. 1 would be a better summary of your paper if it included the relational results as in Fig. 4b (the figures are identical other than this omission). Given the brute-force group method is computationally infeasible, it would be useful to communicate early on the performance of the proposed approximation. Also I would like to see the MATES baseline included in Figure 3.

---

> ### Author Rebuttal · Authors · 2025-07-30
>
> Thank you for your time and insightful review of our paper! We address your questions/comments below:
>
> **Weakness 1:** The gains over MATES are modest and there remains a large gap between Group-MATES and the brute-force group selection. It is unclear whether the additional computational work (and overhead to set up the method) is worthwhile for practitioners over simply using MATES.
>
> **Response:** In the table below, we demonstrate the actual wall clock time for each part of our method, compared with MATES and brute-force selection. The main difference between Group-MATES and MATES lies in the oracle data influence collection step, where Group-MATES collects 20k rollout trajectories (each containing 10 oracle data influences) and MATES collects 80k locally probed oracle data influences. However, this gap only accounts for **less than 3%** of the total wall clock time relative to pretraining hours. Notably, our data selection can be performed solely on CPUs after the data influence model inference, and is highly parallelizable—e.g., different clusters can utilize separate cores for independent selection. Considering the 2x net efficiency gain achieved by Group-MATES compared to MATES (see next response), this additional wall clock time is negligible.
>
> Furthermore, as you noticed, brute-force group selection is computationally infeasible, and that’s why we only use a 100-step training to showcase its performance. Group-MATES effectively approaches brute-force performance at a pretty low additional cost, and we hope our work inspires future research in this promising direction.
>
> |                                               | Group-MATES (400M-4x) | MATES (400M-4x) | Brute-Force Group Selection (400M-4x) | Group-MATES (1B-1x) | MATES (1B-1x) | Group-MATES (3B-1x) | MATES (3B-1x) |
> | --------------------------------------------- | --------------------- | --------------- | ------------------------------------- | ------------------- | ------------- | ------------------- | ------------- |
> | Model Pretraining (H100 Hours)                | 104                   | 104             | 104                                   | 240                 | 240           | 740                 | 740           |
> | Oracle Data Influence Collection (H100 Hours) | 4.5                   | 1.8             | 22632240                              | 12.1                | 4.8           | 18.7                | 7.5           |
> | Data Influence Model Training (H100 Hours)    | 2.7                   | 2.0             | -                                     | 2.7                 | 2.0           | 2.7                 | 2.0           |
> | Data Influence Model Inference (H100 Hours)   | 14.2                  | 14.2            | -                                     | 12.0                | 12.0          | 23.9                | 23.9          |
> | Data Selection (208-Core CPU Time)            | 10min                 | 1min            | 151min                                | 8min                | 1min          | 17min               | 2min          |
>
> ---
>
> **Weakness 2:** MATES baseline in Figure 3.
>
> **Response:** Thank you for your suggestion! Below is the comparison of Group-MATES and MATES performance w.r.t. training tokens. For each method, we report **Core score/total FLOPs (1e19)/total GPU hours** in each cell. “Total” here means we consider all costs from data selection and model pretraining.
>
> *400M-4x (cells used to calculate net efficiency gain are marked italic):*
>
> | Tokens (B)           | Random           | Quad           | MATES           | Group-MATES     |
> | -------------------- | ---------------- | -------------- | --------------- | --------------- |
> | 19.7                 | 0.193/4.8/62     | 0.196/5.8/85   | 0.196/5.1/73    | 0.198/5.2/75    |
> | 26.2                 | 0.208/6.4/83     | 0.216/8.4/114  | 0.214/7.0/98    | *0.222/7.1/100* |
> | 32.8                 | 0.214/8.0/104    | 0.224/11.0/142 | *0.223/8.9/122* | 0.234/9.1/125   |
> | 39.4 (for reference) | 0.218/9.6/125    | -              | -               | -               |
> | 45.9 (for reference) | *0.221/11.2/146* | -              | -               | -               |
>
> *1B-1x (cells used to calculate net efficiency gain are marked italic):*
>
> | Tokens (B)           | Random           | Quad           | MATES            | Group-MATES      |
> | -------------------- | ---------------- | -------------- | ---------------- | ---------------- |
> | 16.8                 | 0.257/14.4/144   | 0.260/15.3/159 | 0.262/14.8/155   | 0.266/15.0/160   |
> | 22.4                 | 0.283/19.2/192   | 0.288/21.0/212 | 0.291/19.9/207   | 0.298/20.3/213   |
> | 28.0                 | 0.295/24.0/240   | 0.293/26.8/265 | *0.303/25.1/259* | *0.308/25.7/267* |
> | 33.6 (for reference) | *0.304/28.8/288* | -              | -                | -                |
> | 39.2 (for reference) | *0.308/33.6/336* | -              | -                | -                |
>
> We can see that Group-MATES achieves **31.5%** and **20.5%** net efficiency gains (measured by GPU hours) in 400M-4x and 1B-1x setups, respectively. By contrast, MATES only achieves **16.4%** and **10.1%** net efficiency gains in 400M-4x and 1B-1x setups. Therefore, Group-MATES nearly **doubles** the net efficiency gain of MATES, further validating the advantage of our group-level selection over individual selection. We will include this comparison in Figure 3 and add a plot with GPU hours as the x-axis.
>
> ---
>
> **Question 1:** The paper says the results in Table 1 are for models pretrained from scratch. At what step in training do you start your data rollouts for training the relational data influence model?
>
> **Response:** In the first 10B tokens, the model of Group-MATES is trained on randomly selected data. We then collect rollouts twice after training on 10B and 20B tokens for the 400M-4x and 1B-1x settings, and after 10B and 30B tokens for the 3B-1x setting. The model is continuously trained on the selected data, following the model-aware data selection pipeline in MATES (L206-L209). We will make this setup clear in our Section 5.
>
> ---
>
> **Question 2:** Did you try different similarity functions or embedding models for the relational data influence model?
>
> **Response:** Yes. We experimented with different similarity functions (cosine, dot product, FFN) and embedding models (BERT vs. BGE) in Appendix C.4. We found that BGE outperforms BERT in approximating oracle data influences due to its optimization for sentence embeddings. Among similarity functions, using a trainable FFN achieves similar performance with our cosine similarity but adds parameters, while using dot product significantly reduces performance. These results validate our choice of cosine similarity that aligns with the original BGE. We will highlight this analysis in our main content and add an overview for Appendix C to better guide the readers.
>
> ---
>
> We sincerely thank you again for your valuable time and thoughtful review. We're delighted that you recognize the motivation of our method and highlight our superior performance. In our revised version, we will compare different methods w.r.t. wall clock time and put all baselines in Figure 3.

---

> > ### Comment · Reviewer_zK2t · 2025-08-06
> > **Response to Rebuttal**
> >
> > I have read the author’s rebuttal and the other reviews. The addition of wall clock time measurements for MATES and Group-MATES is a substantial improvement to the paper and demonstrate to me that the performance gains from data selection will not necessarily be outweighed by the data selection methods. I would encourage the authors in future revisions to include relative efficiency gains in terms of wall clock time in addition to those given in terms of training tokens in Figure 3. I also appreciate the inclusion of the MATES baseline in Figure 3.
> >
> > To reflect the improvements to the paper from the rebuttal, I am raising my score. My overall summary is that the gains from Group-MATES over MATES are modest which is still reflected in my significance score but the revisions from the rebuttal now give the reader a much clearer picture of the computational cost and relative efficiency gains of these methods.

---

> > > ### Author Response · Authors · 2025-08-06
> > >
> > > Thank you very much for recognizing our rebuttal and for your helpful suggestions!
> > >
> > > We will include the efficiency comparison with respect to wall clock time, as well as add the MATES baseline to Figure 3 in the revised version of the paper.

---

> ### Author Response · Authors · 2025-08-06
>
> Dear Reviewer,
>
> Thank you once again for taking the time to review our paper. If there are any additional concerns or questions, please do not hesitate to let us know. We would be happy to address them promptly.
>
> Thank you!

---

### Comment · Area_Chair_FF8r · 2025-08-06

Dear Reviewers,

Thank you for your time and effort in reviewing this paper. If you haven’t done so already, please take a moment to read all other reviews and the author's rebuttals. We also encourage you to engage in further discussion with the authors. If you plan to do so, please post your follow-up questions as soon as possible so that the authors have sufficient time to respond.

Thank you for your continued contributions to the review process.

Best,

AC

---

### Note · Authors · 2025-08-12

We sincerely thank all four reviewers for their thoughtful and constructive feedback. Reviewers highlighted multiple strengths of our work, including a clear motivation and principled method for group-level selection (zK2t, aQV7), comprehensive experiments with detailed ablation studies (aEny, NRhp), and superior performance compared to baselines (zK2t, aEny, aQV7, NRhp). Taken together, we are grateful for the recognition that Group-MATES offers a well-founded, solid, and effective framework for group-level pretraining data selection—an underexplored yet critical direction for efficient large-scale pretraining.

Across reviews, a main concern (zK2t, aEny, NRhp) was the perceived modest gains over MATES and whether they justify the added complexity of Group-MATES. In our rebuttal, we provided a detailed wall-clock breakdown showing that Group-MATES adds **less than 3%** wall clock time compared to MATES, yet **nearly doubles** the net efficiency gain of MATES (31.5% vs. 16.4% in 400M-4x; 20.5% vs. 10.1% in 1B-1x). The efficiency advantage addressed this concern reasonably, making all three reviewers increase their scores.

Another shared point was our evaluation scope that primarily focuses on DCLM (aEny, aQV7). While DCLM is the standardized setup for pretraining data curation, we also demonstrated generalization to the C4 dataset in Appendix C.6, where Group-MATES surpassed or matched MATES on **8 out of 9** tasks. Both reviewers agreed that their concerns have been resolved.

Several reviewers noted presentation issues such as improving Figure 1a/6b readability (aQV7), clarifying notations (NRhp), and reorganizing chunked sections (aQV7). We answered these questions in detail and provided specific revision plans, which are recognized by the reviewers.

In summary, our rebuttal and ensuing discussion clarified computational trade-offs, highlighted our significant net efficiency gain over MATES, and demonstrated generalization of our method, leading most reviewers to raise their scores. In our revised version, we will:

1. Add all baselines in Figure 3 w.r.t. wall clock time (GPU hours), along with a detailed compute breakdown.
2. Enhance guidance to C4 generalization results and key ablation studies.
3. Improve figure clarity, refine notations, and restructure main sections for better flow.

We believe the revised paper will be of better quality and clarity. Thank you again for your valuable time and thoughtful consideration!

---

### Decision · Program_Chairs · 2025-09-17

**Decision:**

Accept (poster)

**Comment:**

This paper studies group-level data selection for efficient language model pretraining. The authors introduce Group-MATES, a method that leverages a separately trained relational data influence model for data selection. The authors also propose a cluster-based approach to further accelerate the selection process. Empirical results demonstrate the effectiveness of the proposed approach. After the rebuttal, all reviewers leaned toward acceptance.